# Convection-permitting climate model representation of severe convective wind gusts and future changes in southeastern Australia

Andrew Brown[*,1,2], Andrew Dowdy[1,2], and Todd P. Lane[1,2]

[1]ARC Centre of Excellence for Climate Extremes, The University of Melbourne, Australia
[2]School of Geography, Earth and Atmospheric Sciences, The University of Melbourne, Australia
[*]andrewb1@student.unimelb.edu.au

**Abstract.**

Previous research has suggested that the frequency and intensity of surface hazards associated with thunderstorms and convection, such as severe convective winds (SCWs), could potentially change in a future climate due to global warming. However, because of the small spatial scales associated with SCWs, they are unresolved in global climate models, and future climate projections are uncertain. Here, we evaluate the representation of SCW events in a convection-permitting climate model (Bureau of Meteorology Regional Projections for Australia, BARPAC-M), run over southeastern Australia for December–February months. We also assess changes in SCW event frequency in a projected future climate for the year 2050, and compare this with an approach based on identifying large-scale environments favourable for SCWs from a regional parent model (BARPA-R). This is done for three different types of SCW events that have been identified in this region, based on clustering of the large-scale environment. Results show that BARPAC-M representation of the extreme daily maximum wind gust distribution is improved relative to the gust distribution simulated by the regional parent model. This is due to the high spatial resolution of BARPAC-M output, as well as partly resolving strong and short-lived gusts associated with convection. However, BARPAC-M significantly overestimates the frequency of simulated SCW events, particularly in environments having steep low-level temperature lapse rates. A future decrease in SCW frequency under steep lapse rate conditions is projected by BARPAC-M, along with less frequently favourable large-scale environments. In contrast, an increase in SCW frequency is projected under high surface moisture conditions, with more frequently favourable large-scale environments. Therefore, overall changes in SCWs for this region remain uncertain, due to different responses between event types, combined with historical model biases.

## 1 Introduction

Damaging surface winds can have large impacts on society, and are important to consider in designing buildings and infrastructure. In the midlatitude regions of southeastern Australia, extreme surface wind speeds (that is, gusts that exceed a 20-year average recurrence interval) tend to be associated with convection (Holmes, 2002), with these events sometimes known as severe convective winds (SCWs). As discussed by Wakimoto (2001), these events can be divided into two broad classes of related physical processes: Firstly, there are convective downdrafts (or "downbursts") associated with precipitation and evaporative cooling that can transport momentum to the surface (Fujita, 1985; Wakimoto, 1985; Srivastava, 1985; Atkins and

Wakimoto, 1991; Geerts, 2001), including in supercell rear flank downdrafts (Klemp and Rotunno, 1983). Secondly, there are mesoscale downdraft processes, such as bow echoes, derechos, and associated rear inflow jets, that can form as a result of organised convection (Johns and Hirt, 1987; Wakimoto et al., 2006). Both classes of processes can sometimes be embedded within synoptic-scale weather systems, with the vertical mixing of strong winds from aloft having a large contribution to severe surface winds in these situations (Ludwig et al., 2015; Pantillon et al., 2020).

Some of these convective processes may possibly be affected by human-induced climate change. For example, global warming is expected to increase surface heat and moisture availability for deep moist convection and severe thunderstorms, based on estimates of large-scale environmental changes from global climate models (e.g., Trapp et al., 2009; Púčik et al., 2017; Lepore et al., 2021) and theoretical changes in the convective environment due to tropospheric warming (Ye et al., 1998; Singh et al., 2017). Future human-induced climate change could also potentially impact other factors relevant for deep moist convection,

such as temperature lapse rates, vertical wind shear, and relative humidity (Seeley and Romps, 2015; Chen et al., 2020). Therefore, the frequency of large-scale environments favourable for SCWs is expected to shift in the future, but with significant uncertainties related to a wide range of associated processes such as those mentioned above, as well as a relatively limited number of regional studies (Martinez-Alvarado et al., 2018; Brown and Dowdy, 2021a; Prein, 2023). Also, these methods for future projections based on the large-scale environment do not consider potential changes in storm-scale processes that may be

sensitive to future warming, such as those related to internal storm dynamics and storm mode, the distribution of hydrometeors, or potential changes in storm initiation mechanisms and convective inhibition (e.g., Hoogewind et al., 2017; Allen, 2018; Raupach et al., 2021). These limitations have resulted in relatively low confidence in future projections of severe thunderstorms and SCWs based on environmental changes (Seneviratne et al., 2021).

To increase confidence in future projections of severe convection, methods based on large-scale environmental changes have

recently been complemented by approaches that use convection-permitting atmospheric models, regionally nested within global climate models. These models often show significant improvements in representing convective hazards relative to coarse-scale models, due to partially resolving deep convection (e.g. as reviewed by Prein, 2015). Recent studies have used these convection-permitting models to estimate changes in severe convection and hailstorms in the United States (Trapp et al., 2011; Gensini and Mote, 2015; Trapp et al., 2019; Ashley et al., 2023) and convective rainfall in Europe (Kendon et al., 2017; Chan et al., 2023)

and Africa (Kendon et al., 2019), for example. These studies have shown that convection-permitting models can elucidate important details in future projected changes that are not resolved by coarse-scale models and methods. For example, projected decreases in the rate of convective initiation have been reported in the United States, but with increases in the intensity of thunderstorms that do occur based on simulated radar reflectivity and updraft speeds (Hoogewind et al., 2017; Haberlie et al., 2022).

A limited number studies have focused on future regional changes in surface wind gusts related to convection using convection-permitting models (Trapp, 2021; Van de Walle et al., 2021; Dowdy et al., 2021; Manning et al., 2023). These studies have suggested potential future increases in the frequency of SCWs, but still with significant uncertainties related to a small number of regions and models. In addition, while convection-permitting models might be able to broadly represent convective downdrafts and internal storm dynamics leading to SCWs, as shown previously for individual cases (Hawbecker et al.,

2017; Bolgiani et al., 2020), there are often significant errors in the timing, location, and characteristics of severe convection in general (Weisman et al., 2008; Kendon et al., 2021). These errors are likely related to several dynamical processes that are not sufficiently resolved by these model configurations, such as microphysical processes and entrainment of environmental air (Bryan and Morrison, 2012; Jucker et al., 2020; Bergemann et al., 2022). Errors in convective storm timing and location have also been shown to be caused by deficiencies in boundary conditions in some cases, relating to representations of the large-scale thermodynamic environment (Hanley and Lean, 2021). Also, the turbulent nature of SCW events means that the relevant surface winds are often parameterised in these models (Hawbecker, 2020), which can lead to biases in surface wind speeds. Further, the extent of the abovementioned biases may vary with different physical SCW processes that can have different spatial scales, including the convective-scale and mesoscale processes outlined above. Many of these physical processes are potentially relevant for SCWs in southeastern Australia, as demonstrated by Brown et al. (2023).

This study will further complement recent developments in climate modelling of severe convection, by examining the representation of SCWs in a convection-permitting modelling framework, in southeastern Australia, compared with coarse-scale models and weather station observations. Future projected changes in simulated SCW events will then be assessed, and compared with future changes based on the large-scale convective environment. Environmental data will also be used to separate different types of SCW events in the model and in observations, to examine potential variations in the quality of model representation and future projected changes between these event types. The paper is structured as follows: Firstly, the modelling framework being examined here is introduced, as well as the observations-based dataset using station wind gust measurements. Secondly, the intensity distribution for all daily maximum wind gusts over the domain of southeastern Australia is compared between the models and observations. Then, the characteristics of convection-associated severe wind gusts are compared between the convection-permitting model and the observed dataset. This includes an analysis of wind gust duration and strength compared to the background flow, the spatial variability of severe events, and variations in these characteristics between different types of SCW events. Finally, future projected changes in simulated SCW events and favourable SCW environments are presented and compared, before a discussion of the results and concluding comments.

## 2 Datasets and methods

### 2.1 Bureau Atmospheric Regional Projections for Australia (BARPA)

The convection-permitting model used here is a part of the broader BARPA modelling framework from the Australian Bureau of Meteorology (Su et al., 2021). BARPA downscales global atmospheric model data to regional domains using the Australian Community Climate and Earth Systems Simulator (ACCESS), which is based on the Global Atmosphere configuration of the UK Met Office Unified Model. Here, BARPA hindcast simulations over 1990–2015 covering two, nested domains are analysed: Firstly, BARPA-R, which is a regional simulation with a domain that covers eastern Australia and secondly, BARPAC-M, which is a convection-permitting simulation nested within BARPA-R with a domain covering southeastern Australia (Figure 1). The BARPAC-M simulations were run after the full time period of BARPA-R simulations were completed. BARPAC-M can therefore be considered an offline simulation with no feedback into the BARPA-R simulation. The BARPA-R simulation used

here is configured with ACCESS at 12 km horizontal grid spacing, downscaled from ERA-Interim (that is on a 0.75-degree latitude-longitude grid, see Dee et al., 2011) at the boundaries with no data assimilation. BARPAC-M is configured with ACCESS on a 2.2 km horizontal grid, and therefore does not parameterise deep convection. The BARPAC-M simulations use a sponge zone for the lateral boundary nesting in the BARPA-R simulations, with analysis excluding data from that sponge zone, as was also the case for the BARPA-R simulations nested in ERA-Interim (Su et al., 2021). The BARPAC-M data available for use here covers the months December–February, over the same years as BARPA-R.

In addition to these hindcast simulations, two 20-year BARPA climate simulations are also analysed. These are downscaled from the ACCESS1-0 global climate model (that is on a 1.25-degree latitude and 1.875-degree longitude grid, see Bi et al., 2013) under a historical forcing scenario over 1985–2005, and an RCP8.5 scenario over 2039–2059. Historical and future climate simulations have the same configurations as mentioned above, for BARPAC-M and BARPA-R.

In each BARPA configuration, surface wind gusts are parameterised by adding a turbulent component to the predicted 10 metre wind speed. This parameterisation is intended to represent a gust defined by 3-second average wind speed, based on the standard deviation of the resolved horizontal wind speed, surface roughness, and the stability of the surface layer (Ma et al., 2018). Parameterised 3-second wind gust output is saved at 10-minute intervals in both models. For BARPAC-M, the 10-minute output represents the maximum 3-second gust over all model time steps. In contrast, for BARPA-R, the 10-minute output represents the wind gust from a single model time step, based on a model time step of 5 minutes. For the purposes of comparing the two model configurations, the BARPA-R 3-second gust distribution based on instantaneous 10-minute output is expected to be similar to a 10-minute maximum, given the relatively coarse model time step of 5 minutes, but with slightly lower mean and extreme values. This is demonstrated in Supplementary Material (Section S1) by resampling observational wind gust data to 5-minute intervals, where a bias of around -1.0 to -0.5 m/s is introduced using 10-minute instantaneous observed gusts relative to 10-minute maximum observed gusts. However, this bias is not expected to significantly impact the analysis of extreme wind gusts associated with convection, where spatial resolution and physical process representation are most relevant when comparing between models (as will be shown in Section 3.1 and 3.2).

Lightning flashes are also estimated by BARPAC-M, and saved as daily output (number of flashes each day). Lightning flashes are parameterised based on an upwards graupel flux approach, combined with the distribution of ice-phase hydrometeors, that is known to correlate with observed lightning (McCaul et al., 2009). Lightning flashes are matched with BARPAC-M and BARPA-R wind gusts using a radius of 50 km on the same day.

## 2.2 Observations of wind gusts and convection

Wind gust observations are obtained from a large number of automatic weather stations (AWS) managed by the Australian Bureau of Meteorology, at 272 locations in southeastern Australia (Figure 1). All available AWS were used here, except for those located more than 1000 m above sea level, and those located offshore. These stations are excluded because of difficulties in identifying convective wind gusts. Wind gusts are reported as the maximum in one-minute intervals, and represent a 3-second average wind speed at a height of 10 meters. The 10-minute maximum of these one minute observations are taken to match with the BARPA hindcast wind gust data. Automatic quality control information is provided with each wind gust measured by

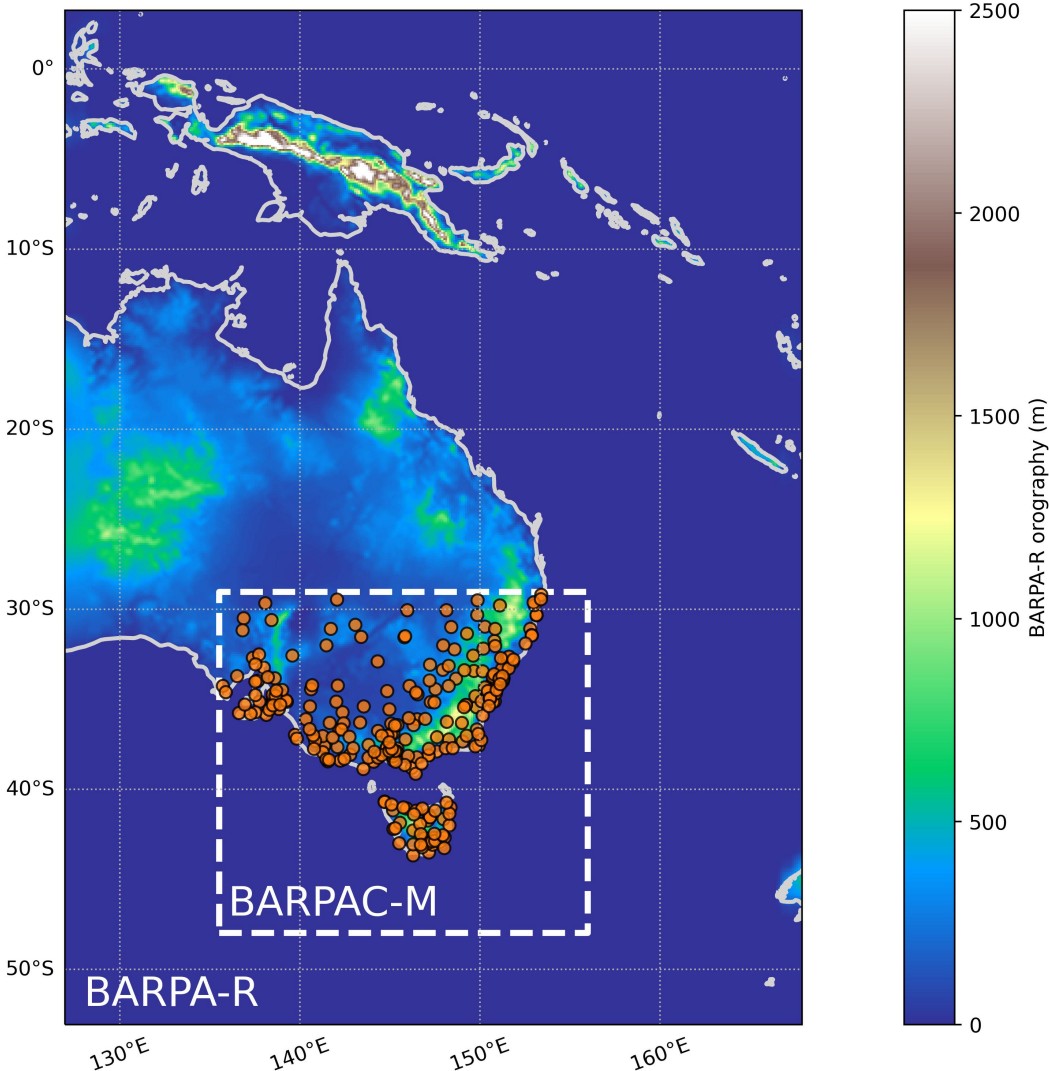

**Figure 1.** BARPA-R domain, with inner BARPAC-M domain indicated by white dashed lines. The location of all automatic weather stations used here are indicated with orange circles. Orographic heights used in BARPA-R are shown with shaded contours.

AWS, with only data that has passed this quality control check retained here. In addition, six instances of erroneously large wind gusts are manually removed based on unrealistic gust evolution and/or gust speeds.

Lightning observations from the World Wide Lightning Locator Network (WWLLN Virts et al., 2013) over a period of 2005–2015 are also analysed in relation to severe wind gusts. Lightning stroke counts are binned onto a 0.25-degree latitude-longitude grid, at hourly intervals. The hourly grid values are then resampled to daily sums, to match with BARPAC-M parameterised lightning output, and grid cells are matched spatially to station locations with a 50 km radius.

## 2.3 Environmental diagnostics

Large-scale diagnostics that describe the convective environment and are relevant for SCWs are calculated, using 6-hourly BARPA-R and ERA5 output on pressure and surface levels. These environmental diagnostics will be used for clustering of different SCW types (see Section 3.3 and Supplementary Material Section S2 for further details), as well as for identifying favourable SCW environments in Section 4. To identify favourable SCW environments, the Brown and Dowdy (2021b) Statistical Diagnostic (BDSD) is applied. The BDSD is based on a logistic regression approach that relates environmental variables to the probability of a measured SCW event, in the form:

$$BDSD = \frac{1}{1 + e^{-z}} \tag{1}$$

where

$$
\begin{aligned}
z = {} & 6.1 \times 10^{-2}(EBWD) + 1.5 \times 10^{-1}(Umean800\text{–}600) + 9.4 \times 10^{-1}(LR13) + 3.9 \times 10^{-2}(RHMin13) \\
& + 1.7 \times 10^{-2}(SRHE) + 3.8 \times 10^{-1}(Q\text{-}melting) + 4.7 \times 10^{-4}(Eff\text{-}LCL) - 1.3 \times 10^{1}
\end{aligned}
\tag{2}
$$

and EBWD is the effective bulk wind difference (m/s), Umean800–600 (m/s) is the mass-weighted mean wind speed over the 800–600 hPa layer (m/s), LR13 is the temperature lapse rate from 1–3 km above ground level (°C/km), RHMin13 (%) is the minimum relative humidity in the 1–3 km layer, SRHE (m$^2$/s$^2$) is the effective storm relative helicity, Q-melting (g/kg) is the water vapor mixing ratio at the height of the melting level, and Eff-LCL (m) is the lifting condensation level of an effective-layer parcel. The effective layer and related quantities are defined following Thompson et al. (2007). Here, following Brown and Dowdy (2021a) a threshold probability of 0.83 is applied to Equation 1 to classify an environment as being favourable. For further details on the definition and calculation of this diagnostic, please refer to Brown and Dowdy (2021a, b).

Each BARPAC-M wind gust is matched to BARPA-R environmental diagnostics using the most recent 6-hourly timestep before the gust, and by taking the maximum of each diagnostic in a 50 km radius around the gust location using model land points only, following Brown et al. (2023). Similarly, each observed gust is matched to ERA5 environmental diagnostics using the same method. It is noted that the distribution of relevant environmental diagnostics between ERA5 and BARPA-R are very similar, providing confidence in the use of these diagnostics from BARPA-R (see Supplementary Material Section S2).

## 3 BARPA hindcast evaluation

### 3.1 Daily maximum wind gust intensity distribution

Here we compare the wind gust intensity distribution between each of the BARPA hindcast model configurations, the forcing model (ERA-Interim), and observations measured from AWS. BARPA wind gusts are first subset to grid points that are spatially nearest to each AWS location, using model land points only. BARPA data is only retained at times when neighbouring AWS

are reporting quality data, over a 2005-2015 period using only December–February months (with this period representing the overlap between BARPAC-M, BARPA-R, and WWLLN). The daily maximum gust at each station location is then calculated, to reduce the number of samples, while preserving strong gusts that are of interest here. Daily maximum ERA-Interim gusts are calculated in the same way as each BARPA model described above, noting that ERA-Interim gusts are available as the maximum in 3-hourly intervals. It should be noted that the observations used here are representative of a single point location, compared with each of the model datasets that are intended to represent a grid cell average. Therefore, some differences between the observed wind gust distribution and model distributions should be expected, including lower model wind speeds for local wind gust events at station locations in general. The effect of model grid spacing will be investigated later in this section for BARPAC-M.

Figure 2a shows a wind gust intensity histogram for all daily maximum gusts from each dataset, with Figures 2b–e showing quantile-quantile plots for each model compared with the observed dataset. ERA-Interim is shown to realistically represent wind gust percentiles up to around 15 m/s, while significantly underestimating percentiles above this. This is likely due to the large grid cell area of this model dataset, as mentioned above in relation to comparing with point observations. The maximum gust produced by ERA-Interim is around 37 m/s, 7 m/s below the maximum observed gust (Table 1). BARPA-R provides an improved gust distribution for the upper tail, with a realistic estimate of wind gust percentiles up to the 99th percentile, equivalent to around 24 m/s (Figure 2d). For percentiles greater than this, BARPA-R tends to underestimate the wind gust speed compared with observations (Table 1). BARPA-R does produce a small number of gusts between 33 and 44 m/s, although these are all related to the same large-scale low pressure system produced by the model on a single day. Therefore, the uncertainties associated with the estimate of the maximum wind gust in BARPA-R are relatively large, demonstrated by dashed lines in Figure 2d.

BARPAC-M tends to overestimate the observed wind gust distribution for most percentiles by around 1–3 m/s (Figure 2b), although with a better representation of the extreme upper tail (above the 99th percentile) compared with BARPA-R. The physical reasons for this improvement are investigated in Section 3.2, by analysing convective wind gusts in the model. Here, we also examine the effects of horizontal grid size on the representation of the daily wind gust distribution, by presenting the distribution from the BARPAC-M hindcast that has been regridded to the BARPA-R grid (using a conservative interpolation approach). Regridding leads to a reduction in the overestimation of gust speed from BARPAC-M for low percentiles, and a slight underestimation of wind gusts relative to observations for percentiles exceeding the 99.9th (above 30 m/s). The regridded BARPAC-M distribution produces higher wind gust percentiles than BARPA-R for the extreme upper tail, closer to observations (Figure 2c, Table 1). This indicates that added value in BARPAC-M is provided by improvements in relatively small-scale resolved processes, in addition to higher-resolution output.

To examine how changes in large-scale forcing could affect the daily maximum wind gust distribution, we also compare the BARPAC-M hindcast with the historical climate simulation driven by the ACCESS1-0 model. This comparison is performed over a common 1990–2005 period, with distributions for the two BARPAC-M simulations shown in Supplementary Material Section S3. Results indicate that differences in the wind gust distribution between the BARPAC-M simulations with different forcings are relatively small compared with differences between the BARPAC-M hindcast and coarser-scale models (Figure

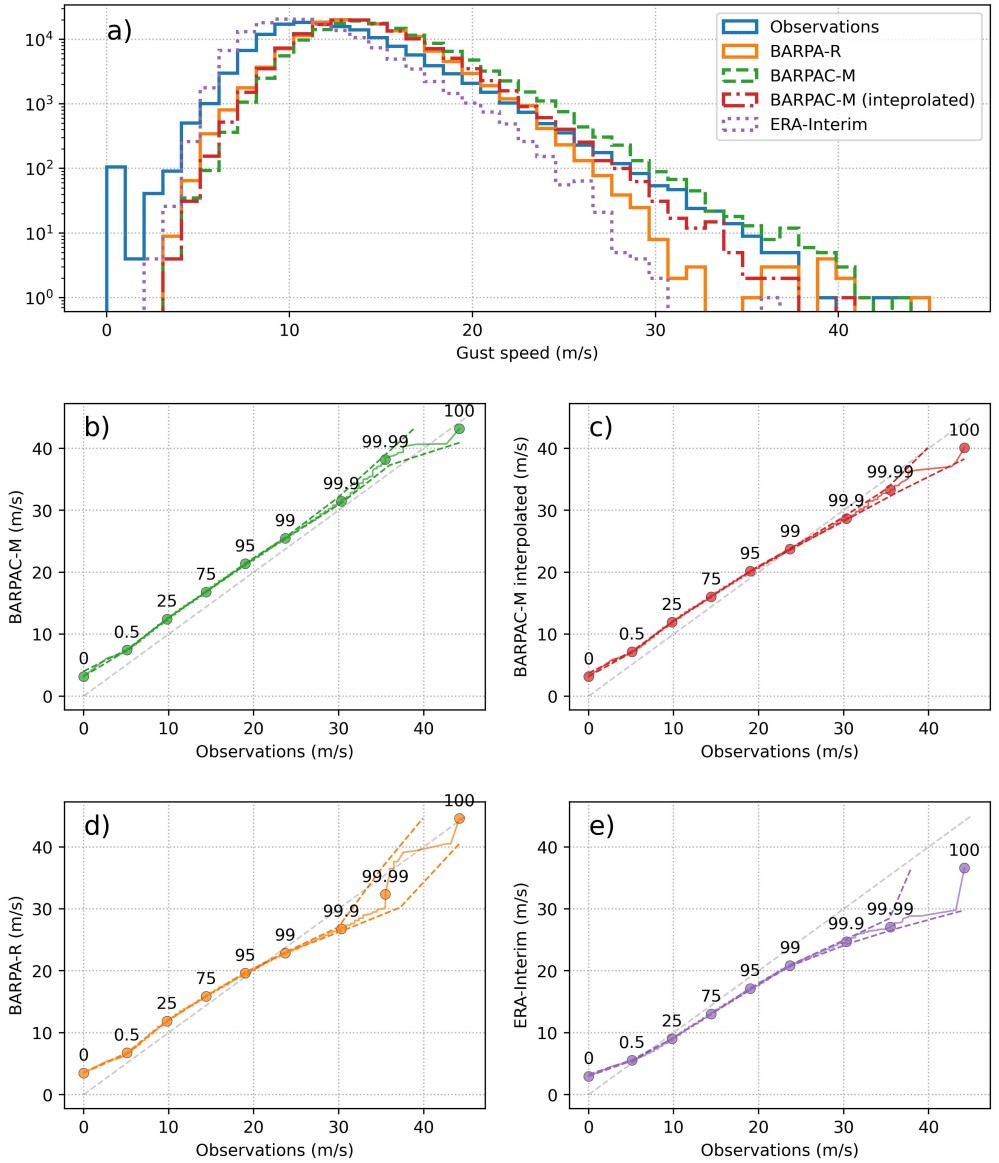

**Figure 2.** a) Histogram of daily maximum wind gust intensity over a 2005-2015 period (December–February months), from BARPA-R and BARPAC-M hindcasts, BARPAC-M hindcast interpolated to the BARPA-R grid, ERA-Interim, and station wind gust observations (with a log scale for counts in each bin). b) Quantile-quantile plot comparing daily maximum BARPAC-M hindcast wind gust percentiles with station wind gust percentiles, with a line plotted using 1,000 evenly spaced percentiles up to the 95th, and then by plotting a number of percentiles equivalent to 5% of the number of samples from the 95th to 100th percentile. A selection of percentiles are also indicated with circle markers, and labelled with text. Dashed lines represents a two-sided 95% confidence interval, calculated by resampling each distribution 1,000 times with replacement, and calculating each of the labelled percentiles. c), d), and e): Same as (b) but comparing BARPAC-M interpolated to the BARPA-R grid, BARPA-R, and ERA-Interim with station wind gusts, respectively.

**Table 1.** Percentile values of daily maximum observed wind gusts (m/s), and the bias in each model dataset for those percentiles. A two-sided 95% confidence interval is shown in parenthesis, estimated by randomly resampling each model distribution 1,000 times, with replacement.

| | Observations (m/s) | BARPAC-M (bias, m/s) | BARPAC-M intepro-lated (bias, m/s) | BARPA-R (bias, m/s) | ERA-Interim (bias, m/s) |
|---|---|---|---|---|---|
| 0th percentile | 0.00 | 3.18 (3.18, 4.03) | 3.18 (3.18, 3.9) | 3.5 (3.5, 3.62) | 2.97 (2.97, 2.97) |
| 0.5th percentile | 5.10 | 2.36 (2.3, 2.42) | 2.07 (2.02, 2.12) | 1.65 (1.53, 1.65) | 0.46 (0.43, 0.5) |
| 25th percentile | 9.80 | 2.64 (2.62, 2.66) | 2.14 (2.12, 2.16) | 2.07 (2.07, 2.07) | -0.79 (-0.81, -0.77) |
| 75th percentile | 14.40 | 2.41 (2.38, 2.44) | 1.63 (1.61, 1.66) | 1.47 (1.35, 1.47) | -1.4 (-1.42, -1.37) |
| 95th percentile | 19.00 | 2.39 (2.33, 2.45) | 1.18 (1.13, 1.23) | 0.62 (0.62, 0.75) | -1.86 (-1.91, -1.81) |
| 99th percentile | 23.70 | 1.79 (1.66, 1.89) | 0.05 (-0.06, 0.14) | -0.82 (-0.95, -0.7) | -2.85 (-2.94, -2.74) |
| 99.9th percentile | 30.31 | 1.16 (0.87, 1.61) | -1.62 (-1.96, -1.45) | -3.56 (-3.81, -3.18) | -5.59 (-5.86, -5.44) |
| 99.99th percentile | 35.50 | 2.71 (1.47, 3.54) | -2.19 (-2.75, -0.88) | -3.14 (-5.64, 2.12) | -8.43 (-8.8, -7.7) |
| 100th percentile | 44.20 | -1.02 (-3.55, -1.02) | -4.1 (-7.65, -4.1) | 0.42 (-4.83, 0.42) | -7.61 (-15.36, -7.61) |

2). However, wind gust percentiles are higher in the historical climate simulation compared with the hindcast simulation, for all percentiles between the 75th and 99.99th. These differences could be due to several factors, including different internal variability within each simulation, or bias in the ACCESS1-0 large-scale environment that could lead to greater atmospheric instability or background wind speeds, for example. Biases in the ACCESS1-0 environment are not examined here, but could be the topic of future work.

**3.2   Wind gust ratio and definition of convective wind gusts**

In addition to improved spatial and temporal resolution of the model output, the improvements in the BARPAC-M representation of extreme wind gusts compared to BARPA-R is likely due to severe gusts produced by convective processes, given that the model is convection permitting (rather than needing to parameterise convective processes). This is evaluated here by using a wind gust ratio approach to associate wind gusts with convection in the BARPA hindcasts and in observations. This
quantity describes the ratio of the daily maximum wind gust to the 4-hour mean at a station location (centred on the time of the gust), calculated based on data at 10-minute intervals. Relatively high wind gust ratios are representative of strong, transient gusts associated with convection, compared with strong gusts generated by synoptic-scale processes that can persist over many hours.

BARPAC-M is able to broadly reproduce the observed wind gust ratio distribution for severe (+25 m/s) daily maximum wind
gusts, while BARPA-R produces much lower wind gust ratio values (Figure 3). Note that the threshold of 25 m/s for severe gusts is chosen for consistency with forecasting definitions (http://www.bom.gov.au/weather-services/severe-weather-knowledge-centre/severethunder.shtml, accessed 22 May 2023), with the number of daily maximum gusts that exceed 25 m/s shown at the bottom of Figure 3. This corresponds to around the 99th percentile based on the entire observational wind gust distribution (Table 1).

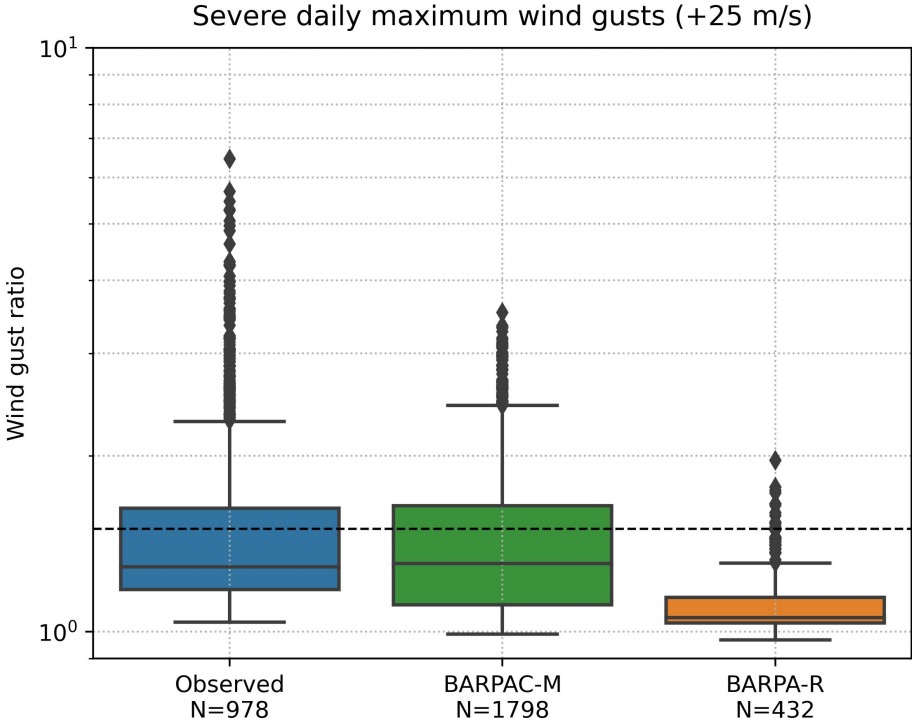

**Figure 3.** Boxplots indicating the distribution of wind gust ratios for severe daily maximum gusts from station observations, BARPAC-M hindcast, and BARPA-R hindcast. Boxes range from the 25th to the 75th percentile, horizontal lines represent the median, and whiskers represent 1.5 times the inter-quartile range. A dashed black line indicates a wind gust ratio of 1.5, and the total number of severe gusts (N) are indicated beneath each boxplot. Note that a logarithmic scale is used on the vertical axis.

A wind gust ratio threshold of 1.5 will be used here to classify a wind gust as convective. This value is consistent with the value used by Durañona et al. (2007) for defining extreme non-synoptic wind gusts, and similar to the value of 2 by El Rafei et al. (2023), noting that the 10-minute maximum gust data used here would be expected to result in lower overall wind gust ratios compared with the one-minute maximum gust data used in that study. A wind gust ratio of 1.5 also appears to reasonably discriminate between lightning-associated daily maximum wind gusts, and non-lightning associated daily maximum wind gusts (Supplementary Material Section S4). This threshold is exceeded by around 31% of severe, daily maximum gusts in observations, and 34% of severe gusts in BARPAC-M, but is only exceeded by 3% of severe gusts in BARPA-R. This is expected given that the BARPA-R model configuration does not have an explicit representation of convective processes, with severe gusts in that model likely driven by larger-scale processes that persist over several hours.

For the remainder of this paper, analyses of simulated and observed wind gusts associated with convection will use a wind gust ratio threshold of 1.5 applied to daily maximum wind gusts. Daily maximum gusts that exceed a wind gust ratio of 1.5 will be referred to as convective gusts, and will be compared with non-convective gusts. By discriminating between two types

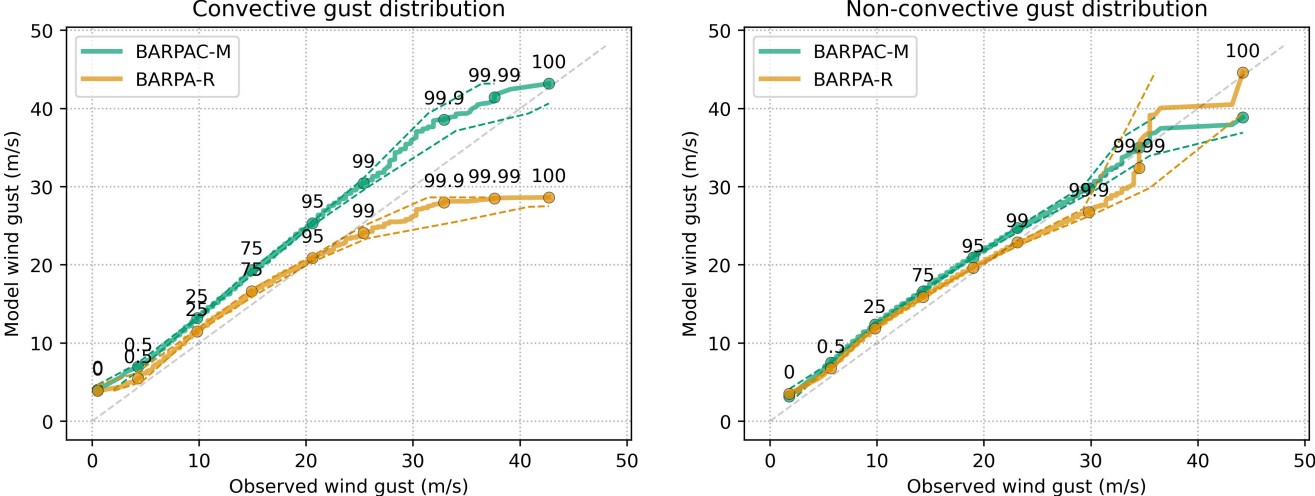

**Figure 4.** As in Figure 2b and d, but for (left) convective daily maximum wind gusts and (right) non-convective daily maximum wind gusts from BARPAC-M (green) and BARPA-R (orange). Convective and non-convective gusts are defined by a wind gust ratio threshold of 1.5.

of wind gusts and comparing each BARPA distribution with observations, improvements in the BARPAC-M extreme wind speed representation can clearly be attributed to the representation of convective gusts (Figure 4). In contrast, the biases in non-convective gust speeds are very similar between BARPAC-M and BARPA-R.

Manual checks of two individual severe convective wind events produced by the BARPAC-M hindcast show that the model
has physically realistic behaviour compared with analogous events based on observations (see Supplementary Material Section S5). This includes spatial wind gust structures that might be expected in reality based on observed reflectivity structures, and simulated wind gust time series' that appear similar to the evolution of analogous measured gusts. This gives confidence in the ability of BARPAC-M to represent severe wind gusts associated with convection, in addition to the statistical approach presented in this section.

**3.3   Different types of severe convective wind events**

Here, we compare characteristics of SCWs in the BARPAC-M hindcast with observations, separately for different types of SCW events. Characteristics include the wind gust ratio, the ratio of the wind gust to the 0–6 km mean wind speed ("deep-layer wind ratio"), wind gust intensity, daily lightning counts, and spatial variability in SCW occurrence frequency accross station locations. For the deep-layer wind ratio, the 0–6 km mean wind speed is calculated from ERA5 for observed events,
and from BARPA-R for the BARPAC-M events. High values of the deep-layer wind ratio represent gusts that are stronger than the background wind flow, indicating that those gusts are generated by internal storm processes, rather than by vertical mixing of strong synoptic winds due to convection, for example. A 0–6 km layer is chosen as this is likely representative of

the background flow relevant for vertical mixing by downdrafts and storm motion, with wind speed data over this layer already available based on its application for event clustering (see Supplementary Material Section S2).

The separation of different event types is important given that there are a range of physical processes that can lead to severe surface winds associated with convection (e.g. Wakimoto, 2001), and the extent to which these processes are represented by BARPAC-M may vary. Three event types are defined here, based on clustering developed by Brown et al. (2023), that depends on the large-scale environment (see Supplementary Material Section S2 for more details). This clustering results in *strong background wind*, *steep lapse rate*, and *high moisture* SCW events. Brown et al. (2023) suggest that strong background

wind events are related to the vertical transport of strong synoptic-scale winds by relatively shallow convection to the surface, while steep lapse rate events appear to be driven by downdrafts associated with the evaporation and melting of precipitation in convective clouds, and high moisture events appear to be associated with the outflow of intense deep moist convection, including supercells. Environmental diagnostics for clustering are calculated using ERA5 for measured events, and BARPA-R for BARPAC-M events, using methods described in Section 2.

Consistent with Brown et al. (2023), Figures 5c, e, and g demonstrate that the observed wind gust ratio, daily lightning flashes, and deep-layer wind ratio all increase with clusters that are supportive of relatively deep convection. That is, these quantities are highest for SCW events in the high moisture cluster, followed by the steep lapse rate and strong background wind clusters. BARPAC-M is able to replicate this behaviour for the distribution of daily lightning flashes (Figures 5f) and the deep-layer wind ratio (Figures 5h). This broad separation of SCW characteristics between different types of events gives

some confidence in the suitability of applying the clustering method to BARPAC-M. In addition, the distribution of large-scale environmental diagnostics is broadly consistent between observations and BARPAC-M across event types, as shown in the Supplementary Material (Section S2). However, there are also some key differences in SCW characteristics between BARPAC-M and the observed dataset, for different event types. For example, while BARPAC-M produces higher wind gust ratios for SCWs associated with the steep lapse rate cluster compared with the strong background wind cluster, as observed

(Figure 5d), the model produces significantly lower values for the high moisture cluster compared with observations. This suggests that key processes related to SCWs within the high moisture cluster are not represented in BARPAC-M. This could relate to, for example, supercell downdrafts that occur on small spatial scales, and mostly occur within this cluster (Brown et al., 2023). There appears to be little difference in the severe wind gust intensity distribution between clusters, based on the results here (Figures 5a–b).

BARPAC-M also appears to broadly reproduce the observed spatial variability of SCW event occurrence frequency between different environmental clusters (Figure 6). This includes strong background wind events mostly occurring in the south of the domain, steep lapse rate events occurring at a wide range of locations including inland, and high moisture events occurring mostly in the coastal and eastern part of the domain. Figure 6 also shows that the total number of events between BARPAC-M and observations is comparable for the strong background wind cluster (100 and 89 from observations and BARPAC-M, re-

spectively) and high moisture cluster (99 and 124), although the amount of steep lapse rate events is significantly overestimated (100 and 401).

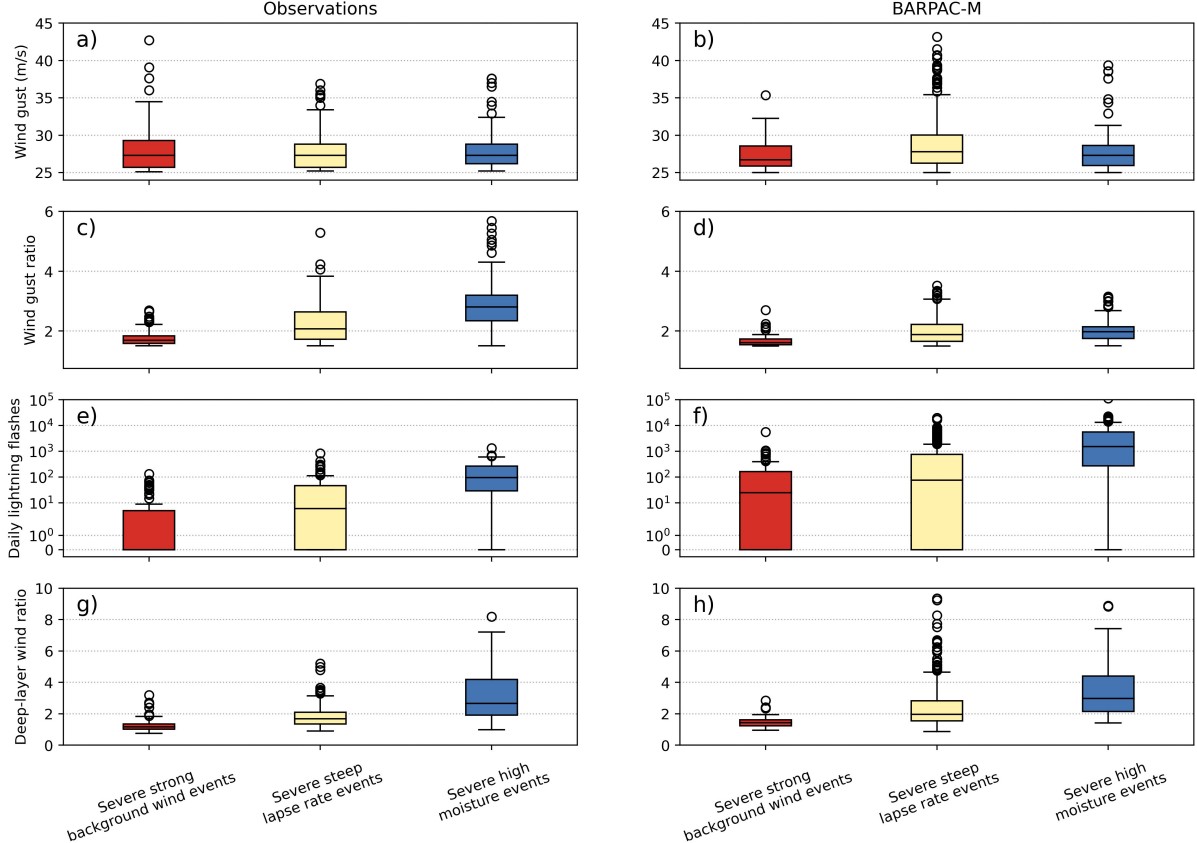

**Figure 5.** Boxplot distributions showing the distributions of (a,b) wind gust speed, (c,d) wind gust ratio, (e,f) daily lightning flashes, and (g,h) deep-layer wind ratio, for severe convective daily maximum wind gusts from (a,c,e,g) station observations and (b,d,f,h) BARPAC-M hindcast. Distributions are shown separately for severe convective daily maximum wind gusts that are associated with: (red boxes) strong background wind, (yellow boxes) steep lapse rate, and (blue boxes) high moisture environments. Boxes range from the 25th to 75th percentile, and whiskers show 1.5 times the inter-quartile range, with circles representing data outside this range

This bias in the number of BARPAC-M SCW events is investigated further here using quantile-quantile plots comparing the daily maximum wind gust distribution between BARPAC-M and observations for each environmental cluster, separately for all convective and non-convective gusts (Figure 7). Results indicate that BARPAC-M tends to overestimate the intensity
distribution of convective daily maximum wind gusts (Figure 7d–f), especially when considering severe gusts over 25 m/s in the steep lapse rate cluster (Figure 7e). This bias results in a greater portion of the BARPAC-M convective wind gust distribution exceeding the 25 m/s severe threshold in steep lapse rate environments, compared with observations, and therefore relates to the overestimation of simulated steep lapse rate SCW events presented earlier (Figure 6c and g) and in the overall number of SCW events (Figure 6a and e). These errors appear to be greatly reduced for non-convective daily maximum wind gusts, but
with a slight high bias (Figure 7a–c).

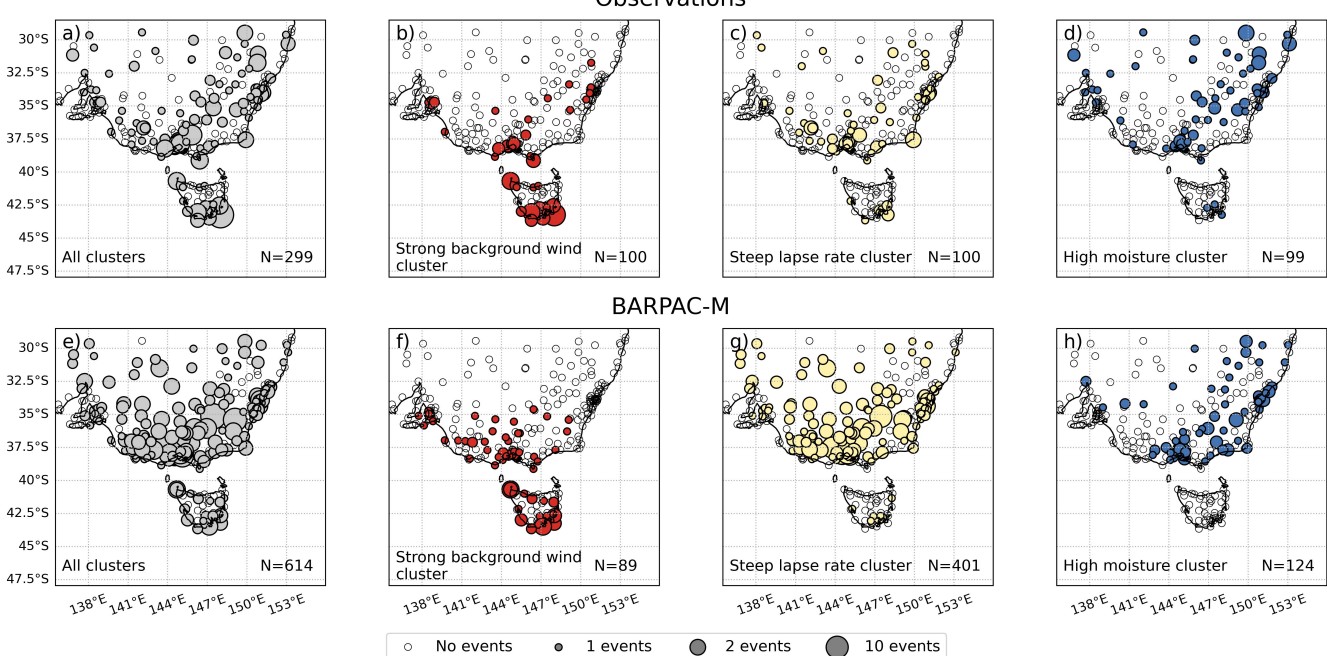

**Figure 6.** Maps showing the relative number of severe convective daily maximum wind gusts at each AWS location, with the size of circle location markers scaled by event occurrences based on (a–d) station observations, and (e–h) BARPAC-M hindcast. Maps are shown separately for (a,e) all environments, (b,f) strong background wind environments, (c,g) steep lapse rate environments, and (d,h) high moisture environments. The total number of events shown on each map is indicated in the bottom-right of each panel. Empty circles represent stations with no events.

This overestimation in the amount of steep lapse rate SCW events could be due to several factors. This includes biases in the large-scale environment inherited from BARPA-R, with a slightly higher occurrence frequency of favourable steep lapse rate environments compared with ERA5 (see Supplementary Material Figure S8b). However, the relative bias in favourable steep lapse rate environments is much smaller than the relative bias in simulated SCW event frequency from BARPAC-M within steep lapse rate environments (compare Supplementary Material Figures S8b and c). This suggests that the bias in simulated SCW event frequency is not primarily driven by biases from BARPA-R, and is instead due to errors in dynamical processes related to SCWs in BARPAC-M, such as convective downdrafts that are too intense or numerous, or errors in gust parameterisations (explored further in Discussion section). This is further supported by a relatively consistent frequency in daily lightning occurrence between BARPAC-M and WWLLN for steep lapse rate environments, compared with the frequency of SCW occurrence (see Supplementary Material Figure S8d).

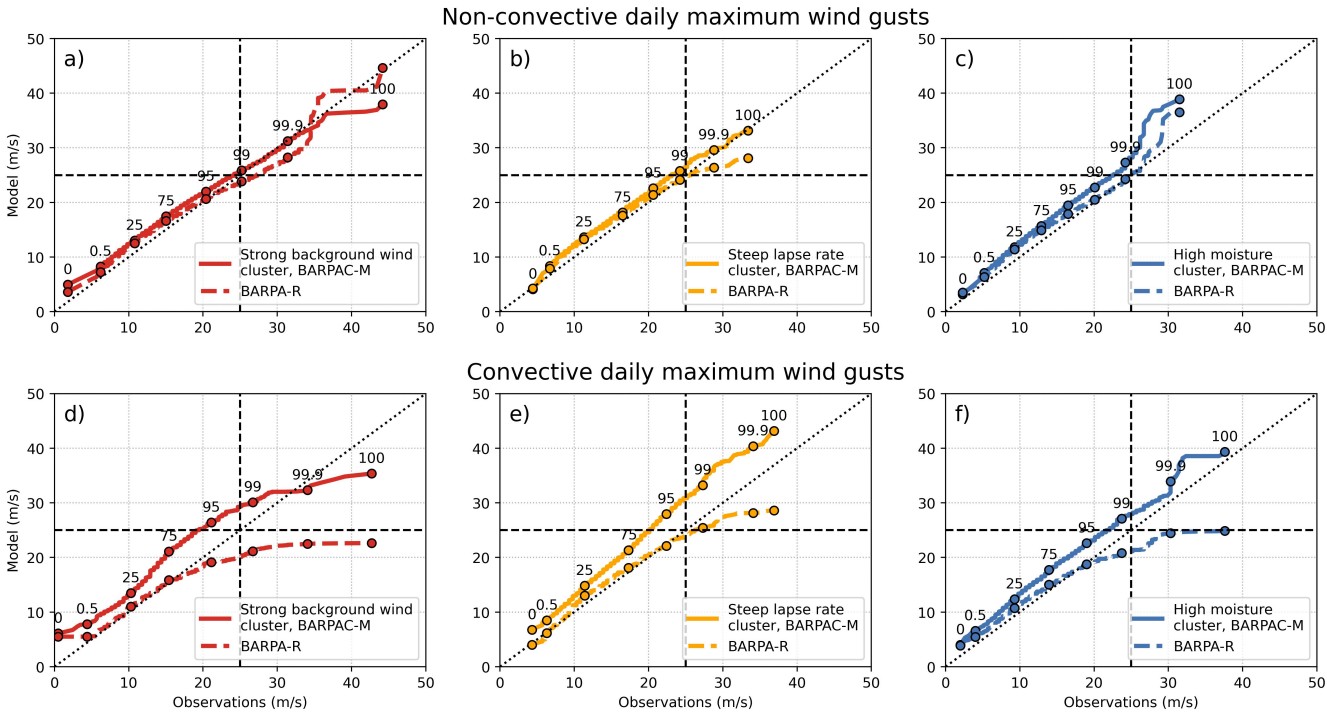

**Figure 7.** Quantile-quantile plots comparing the distribution of daily maximum (a–c) convective gusts and (d–f) non-convective gusts from station observations and BARPAC-M hindcast. Shown for three different environmental clusters: (a, d) strong background wind, (b, e) steep lapse rate, and (c, f) high moisture. Bold dashed lines indicate the threshold used to define severe events. BARPA-R distributions are also shown in a dotted line, for reference. The number of percentiles used for plotting and the labelled percentiles are the same as in Figure 2.

## 4  Future changes in severe convective wind events

Firstly, we compare the entire distribution of daily maximum convective wind gust speeds between historical and mid-century BARPAC-M simulations, forced by the ACCESS1-0 climate model (see Section 2.1). This is done using daily maximum convective wind gusts from the entire model domain, using land points only. According to the results shown in Figure 8,

BARPAC-M suggests an upwards shift in the daily maximum convective wind gust distribution for the extreme upper tail (over 40 m/s), consistent with Dowdy et al. (2021) who also analyse future BARPAC-M wind gust changes (their Figure 4.4c). However, resampling different years in each simulation indicates high levels of uncertainty, with the future wind gust distribution potentially not significantly different from the historical distribution (confidence interval shown by dashed lines in the Figure 8b quantile-quantile plot). Some of this uncertainty is due to extreme gusts that are in some cases related to the same

convective systems (or neighbouring convective systems). This uncertainty is supported by spatial wind gust intensity changes for the extreme upper tail, that have no coherent spatial structure (see 20-year maximum wind gust changes in Supplementary Material Section S7). The spatial changes likely relate to individual convective storm tracks, and result in a low signal to noise ratio. The above points highlight the challenge of assessing future potential changes in the intensity of convective phenomena.

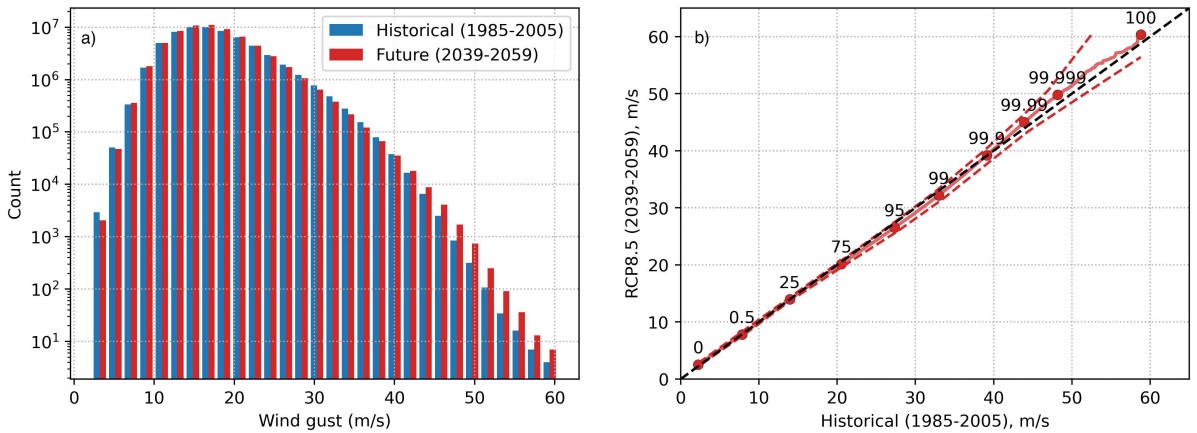

**Figure 8.** (a) Histograms comparing the December–February daily maximum convective wind gust distribution between the historical and RCP8.5 mid-century BARPAC-M simulations, over the entire BARPAC-M domain. (b) Quantile-quantile plot as in Figure 2, but comparing the historical and mid-century convective wind gust distributions, with dashed lines representing a 95% confidence interval calculated by randomly resampling different years from the future BARPAC-M simulation, 1000 times with replacement. The diagonal dashed line represents a theoretical plot if both distributions were identical.

We now investigate future changes in the frequency of SCWs, defined by a wind gust threshold of 25 m/s and wind gust

ratio of 1.5. This is done by calculating the mean daily occurrence probability of simulated SCW events in historical and mid-century BARPAC-M climate simulations, across all spatial (land) grid points. Changes in simulated SCWs will be analysed for each of the event types presented in Section 3.3, with event clustering performed using BARPA-R data. Future changes in simulated SCWs are also compared to changes in the mean daily occurrence probability of a favourable SCW environment

**Table 2.** Mean daily occurrence probability of simulated severe convective wind events from BARPAC-M (SCW) and favourable SCW environments from BARPA-R (F_ENV) for each environmental cluster and all clusters combined. Shown separately for the historical and mid-century climate simulations.

|  | Historical (SCW) | Mid-century (SCW) | Historical (F_ENV) | Mid-century (F_ENV) |
|---|---|---|---|---|
| All events | 0.0257 | 0.0230 | 0.1251 | 0.1387 |
| Strong background wind events | 0.0014 | 0.0008 | 0.0233 | 0.0158 |
| Steep lapse rate events | 0.0206 | 0.0162 | 0.0642 | 0.0589 |
| High moisture events | 0.0041 | 0.0065 | 0.0547 | 0.0820 |

from BARPA-R (F_ENV), using the method of Brown and Dowdy (2021a), as described in Section 2.3. For comparison here,
BARPAC-M simulated SCWs are regridded to the BARPA-R grid, by taking a maximum of one simulated SCW occurrence within each BARPA-R grid cell.

BARPAC-M suggests a decrease in the mean daily occurrence probability of simulated SCWs in the mid-century climate scenario, relative to the historical scenario (Figure 9a), equivalent to a -10.5% change. This decrease is primarily due to decreases in the occurrence probability of steep lapse rate events, given the relatively high base probability of steep lapse
rate events compared with other clusters (Table 2, Figure 6). However, it should be noted that there is a bias for BARPAC-M to produce an unrealistically high number of SCWs in steep lapse rate environments, based on comparisons with observed event frequency (Section 3.3), and this result should be therefore treated with caution. Figure 9a also shows small decreases in simulated SCW probabilities within the strong background wind cluster, while increases are shown for the high moisture cluster. Due to large variability in event occurrences (see spatial maps of change in Supplementary Material Section S7), total
changes are not statistically distinct from zero, although cluster-wise results are less uncertain (95% uncertainty range based on random resampling of different years is shown in Figure 9a). These cluster-wise future changes are similar when considering changes in SCW probability conditional on the occurrence of each cluster (hatched bars in Figure 9a), although the change in conditional occurrence probability for the steep lapse rate cluster may not be statistically distinct from zero, due to the large variability noted previously.

In contrast to overall decreases in SCW probability shown by BARPAC-M, BARPA-R suggests future increases in the mean daily occurrence probability of favourable SCW environments (F_ENVs, see Figure 9b), equivalent to an increase of 10.9%. Although changes in F_ENV for each environmental cluster are the same sign as the changes in simulated SCWs from BARPAC-M, there is a higher increase in high moisture F_ENV probability relative to other clusters, and a smaller relative decrease in steep lapse rate F_ENV probability. This leads to an overall increase in F_ENVs. Considering conditional proba-
bilities, results for the strong background wind and high moisture cluster are the same as for the non-conditional probabilities. However, there is an increase in the probability of a favourable steep lapse rate environment conditional on the occurrence of that cluster, compared with a decrease in the non-conditional probability. This indicates that although there is a decrease in the number of steep lapse rate environments, these are more favourable for SCWs when they do occur. However, the opposite is

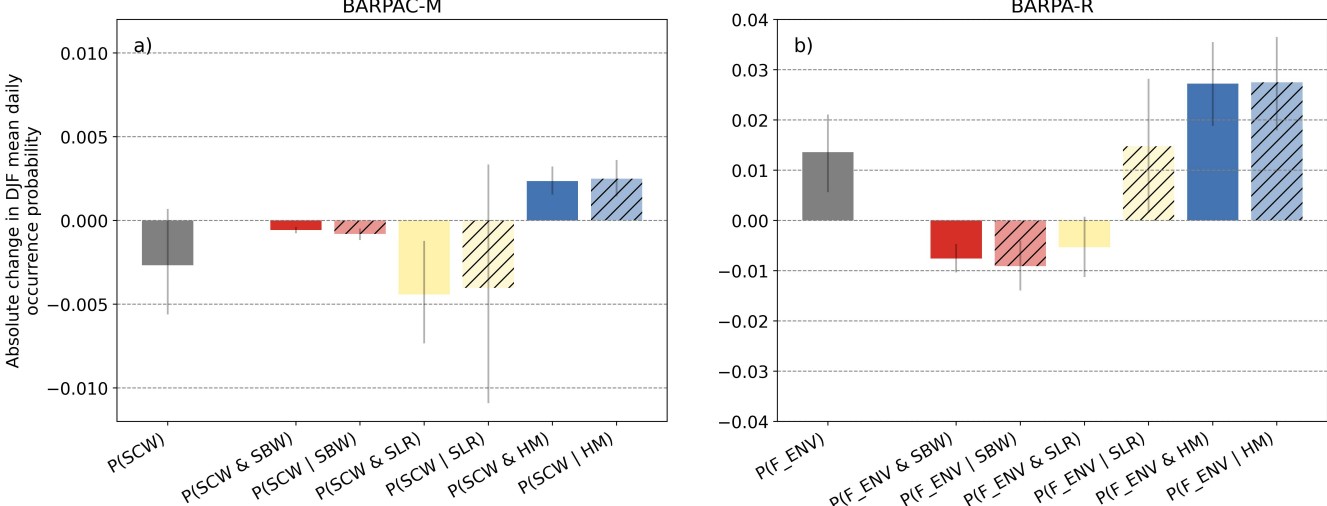

**Figure 9.** a) Change in mean December–February daily occurrence probability of a simulated severe convective wind event (SCW), from BARPAC-M, between the historical and mid-century climate simulations. b) Change in mean daily occurrence probability of a favourable severe convective wind environment (F_ENV), from BARPA-R. Shown separately for strong background wind (SBW, red), steep lapse rate (SLR, yellow), and high moisture (HM, blue) environments, as well as for all events (grey). For individual clusters, two probabilities are shown: the absolute change in the probability of occurrence considering all days (in solid bars, P(SCW & SBW), for example), and the probability conditional on that environmental cluster occurring (hatched bars, P(SCW | SBW), for example). A two-sided 95% confidence interval is shown with grey lines, estimated by randomly resampling years from the future period, 1,000 times with replacement.

true for the conditional probability of simulated BARPAC-M SCWs in steep lapse rate environments, noting large uncertainties
(Figure 9a).

Because there is an overall decrease in SCW probability and an overall increase in F_ENV probability as discussed above, the relationship between the annual number of simulated SCWs and F_ENVs is less correlated in the future period relative to the historical period (Table 3). The relationship based on monthly anomalies also becomes weaker, but to a lesser extent compared with the annual anomalies. The weakening of this relationship potentially highlights the limitation of applying environmental
diagnostics to future projections, given that other factors influencing the formation of severe convection could change in the future, as noted by previous studies (for example Hoogewind et al., 2017).

## 5 Discussion

A convection-permitting configuration of the BARPA modelling framework with 2.2 km horizontal grid spacing (BARPAC-M) shows significant improvements over a 12 km BARPA configuration (BARPA-R) and a global reanalysis that provides
atmospheric forcing (ERA-Interim), in representing the extreme upper tail of the observed daily maximum wind gust distribution over 11 December–February periods. These improvements are shown here to be due to a combination of higher spatial

**Table 3.** Pearson correlation coefficients describing the relationship between the mean annual and monthly anomalies of December–February simulated severe convective wind occurrence from BARPAC-M, and the anomalies of favourable environment occurrences from BARPA-R. Shown separately for the historical and mid-century climate scenarios. A 95% confidence interval is indicated in parenthesis, calculated by randomly resampling the annual and monthly time series 1,000 times with replacement.

|  | Historical | Mid-century |
|---|---|---|
| Annual anomalies | 0.565 (0.202, 0.785) | 0.042 (-0.365, 0.440) |
| Monthly anomalies | 0.613 (0.346, 0.816) | 0.347 (-0.08, 0.691) |

resolution in BARPAC-M output, as well as enhanced representation of severe wind gusts related to convective processes, that are not represented explicitly in coarse-scale models. Findings shown here for extreme winds including severe convective winds (SCWs) in eastern Australia are consistent with numerous studies that have demonstrated the potential added value from convection-permitting models over coarse-scale models, related to convective storm forecasting (e.g. Done et al., 2004; Clark et al., 2009), and the representation of extreme sub-daily precipitation (see reviews by Prein, 2015; Kendon et al., 2021) and extreme wind climates (Manning et al., 2022; Belušić Vozila et al., 2023).

However, BARPAC-M still has significant biases related to the representation of the extreme regional wind climate. This includes an overestimation of SCW event frequency based on the occurrence of convective gusts of at least 25 m/s, particularly in steep lapse rate environments where event frequency is overestimated by a factor of four. This could potentially be related to biases in the parameterisation of turbulent gusts at the surface, that depends on temporal fluctuations in the resolved wind and the surface layer stability profile, and/or the representation of convective processes and storm-scale dynamics. For example, El Rafei et al. (2023) found a systematic positive bias in the turbulent gust representation of a reanalysis version of BARPAC-M, run at 1.5 km grid spacing with data assimilation over subtropical eastern Australia. In addition, bias for too many intense convective cells has been shown previously for the Unified Model (that BARPA is based on) compared with radar observations in the United Kingdom (Hanley et al., 2015) and in tropical Australia (Jucker et al., 2020; Bergemann et al., 2022), and is consistent with the results here for SCW frequency. Unfortunately, we are unable to diagnose further details of the convection and storm dynamics in BARPAC-M here, due to a lack of available data, although this should be a focus of future work.

With these biases in mind, future simulated changes in SCW event probability are analysed from a mid-century RCP8.5 BARPAC-M climate simulation, and compared with changes in favourable SCW environments from the regional BARPA-R model. This comparison is intended to provide insights on the consistency between two commonly used methods for future climate projections of convective hazards. In addition, future changes are compared separately for three types of SCW events that have been shown to occur in this region, based on statistical clustering of the large-scale environment (Brown et al., 2023). A future projected increase in high moisture environments favourable for SCWs is suggested by BARPA-R. These types of favourable environments are often associated with high amounts of CAPE (Brown et al., 2023), and so these future changes agree with previous studies that diagnose hazardous convective environment from GCMs using CAPE-based diagnostics (for example, Allen et al., 2014; Seeley and Romps, 2015; Lepore et al., 2021). In contrast, favourable steep lapse rate environments,

that are likely associated with dry microburst processes (Brown et al., 2023), are projected to decrease in frequency. This supports the findings of Brown and Dowdy (2021a), who showed that the frequency of favourable SCW environments decreases
under an end-of-century RCP8.5 scenario when applying environmental proxies based on temperature lapse rates, compared with an increase when using proxies based on CAPE. In addition, favourable strong background wind environments, that are associated with low CAPE and high amounts of vertical wind shear (Sherburn and Parker, 2014; Brown et al., 2023), are also projected to decrease in frequency. Similar to previous studies in other regions such as Hoogewind et al. (2017) and Haberlie et al. (2022), the BARPAC-M convection-permitting model indicates future changes that are the same sign as the environmental
changes from the driving model, within each type of environment. However, because the majority of BARPAC-M events occur in steep lapse rate environments, the overall future change suggested by that model is a decrease in simulated SCW frequency, while the opposite is true for the BARPA-R environmental method where high moisture environments are relatively important. This indicates that considering different types of events is critical for understanding future projections of SCWs in this region. This also highlights that process-based biases in the frequency of simulated events could potentially have an impact on future
projections, such as an unrealistically high number of steep lapse rate events in BARPAC-M, and that these projections should be treated with caution.

The annual and monthly (December–February) correlation between the number of favourable SCW environments (from BARPA-R) and simulated SCWs (from BARPAC-M) across the entire domain has also been investigated in the historical and future climate periods. In the historical period, an annual Pearson correlation of r=0.565 is reported with a monthly
correlation of r=0.613. This is of a similar magnitude to the correlation between annual SCW environments and measured SCW events around Australia reported by Brown and Dowdy (2021a, r=0.454), and between annual tornado reports and favourable environments in the United States documented by Gensini and Brooks (2018, r=0.66), but lower than the relationship between monthly mean simulated hazardous convection (strong simulated updrafts) and favourable environments reported for the United States by Hoogewind et al. (2017, r=0.91). In the future climate, a much weaker annual (r=0.042) and monthly
(r=0.347) correlation between favourable SCW environments and simulated SCWs is reported here, due to the different future changes in favourable environments and simulated SCWs discussed above. This is in contrast to Hoogewind et al. (2017), who found that this correlation was equally strong in a future climate scenario (r=0.93). The lack of correlation between future simulated SCWs and favourable SCW environments found here highlights potential uncertainties in the application of methods that have been developed in the historical climate, as also mentioned by previous studies (Hoogewind et al., 2017; Raupach
et al., 2021). This includes the application of the Brown and Dowdy (2021b) Statistical Diagnostic (BDSD) that is used here to diagnose favourable SCW environments (see Section 2.3). Additional uncertainty is also potentially introduced here by the fact that the BDSD was initially trained on an observational dataset (Brown and Dowdy, 2021b), and is being applied here to a regional climate model with a potentially different relationship between SCW events and the large-scale environment. Future work could be aimed at further investigating the physical mechanisms relating favourable SCW environments to simulated
events in current and future climates. Although the future changes reported here have low confidence based on this discussion, our results suggest a potential increase in the frequency of SCWs in high moisture environments, that may be compensated by a reduction in SCWs in steep lapse rate environments.

A potential benefit of convection-permitting climate modelling is that simulated changes in the intensity of convective hazards are able to be analysed, compared with environmental approaches that are generally limited to changes in occurrence frequency. Future changes in extreme convective wind gust speeds are relevant for design and adaptation (Holmes, 2002; Lombardo and Zickar, 2020), and have therefore been analysed here using BARPAC-M for southeastern Australia. Although the mid-century BARPAC-M scenario does suggest increases in the occurrence of daily maximum convective gusts that exceed 40 m/s, these changes are not statistically significant based on random resampling of different years. This is due to the high amounts of spatial and temporal variability in these convective-scale events, including many events occurring on a single day in the future period, for example. This demonstrates the challenges of assessing future changes in simulated convective hazards for relatively rare events, such as gusts exceeding 40 m/s. The approach used in this study could be complemented by future work using idealised and/or realistic modelling of projected future individual events, similar to work already underway using pseudo global warming approaches (González-Alemán et al., 2023; Li et al., 2023).

The uncertainties related to extreme wind events mentioned above could also be associated with internal climate variability, including natural modes of variability such as the El Niño-Southern Oscillation (ENSO) and others, that could potentially affect the future projections presented here more generally. Internal variability can be a source of uncertainties in future climate projections (Deser et al., 2012) as well as in historical trends of severe thunderstorm environments for this region (Allen and Karoly, 2014). Uncertainties from internal variability could be exacerbated by the relatively short 20-year period used for the analysis here, noting that this was the maximum period of data available for use and that these data were used in previous research such as described in Dowdy et al. (2021). Simulating long periods with convection-permitting models is very computationally demanding, as noted by previous studies that have used temporal windows of about 10-15 years length for future projections of severe convection (Gensini and Mote, 2015; Ashley et al., 2023). Future work to provide convection-permitting climate model simulations over longer periods will be beneficial, including with a larger sample helping to reduce the influence of internal climate variability (associated with ENSO, for example) on estimates of longer-term climate changes. In addition, the relationships between SCW events and individual modes of climate variability in this region are relatively uncertain, with conflicting results for severe convection based on lightning and hail observations, and severe thunderstorm environments (Allen and Karoly, 2014; Dowdy, 2016; Soderholm et al., 2017; Dowdy, 2020). Future work towards revealing these relationships could likely provide additional insights on the potential impact of internal climate variability on historical and future trends in convective hazards, including severe wind gusts.

## 6    Conclusions

Here, the wind gust intensity distribution from a convection-permitting model hindcast (BARPAC-M) has been evaluated using an observational dataset, and compared with distributions from the associated forcing models. Daily maximum wind gusts are assessed during December–February in southeastern Australia, over an 11-year period. The characteristics of severe gusts associated with convection, as represented by the convection-permitting model, have been compared and contrasted with observed characteristics. This is done for different types of SCW events, as diagnosed by statistical clustering applied to the large-scale environment. Severe convective wind (SCW) event occurrences are then compared between a historical and mid-century climate simulation using BARPAC-M, and future changes in simulated events are compared with changes in environmental conditions that are favourable for SCWs from the regional model that forces BARPAC-M (BARPA-R). Key findings are as follows:

- BARPAC-M shows significant improvements in representing the extreme upper tail (above the 99th percentile) of the observed daily maximum wind gust intensity distribution compared with BARPA-R. Both of these models show improvements over the global forcing model, ERA-Interim, above the 75th percentile.

- Consistent with observations, BARPAC-M is shown to represent severe, transient wind gusts with high wind gust ratios. These gusts are likely related to deep convection and associated outflow, and are not represented by BARPA-R, which parameterises convective processes.

- BARPAC-M can replicate some of the variability in wind gust characteristics and spatial patterns of occurrences between different types of severe convective wind events. However, some key processes appear to not be represented, as evidenced for example by lower wind gust ratios than observed for events in high moisture environments.

- BARPAC-M tends to overestimate the frequency of severe convective wind gusts, particularly in environments with steep lapse rates. Based on findings from previous studies, this bias is likely due to a combination of errors in the surface wind gust parameterisation for convective outflow, as well as in the frequency and/or intensity of deep convection

- BARPAC-M indicates a decrease in the occurrence of simulated severe convective wind events under a mid-century RCP8.5 scenario, driven by a decrease in events within steep lapse rate environments. However, this result is uncertain due to large variability in event occurrences and model biases, as are future estimated changes in convective wind gust intensity.

- In contrast, an overall future increase in the frequency of favourable SCW environments is suggested, driven by an increase in favourable high moisture environments. As a result, the relationship between the annual and monthly number of simulated severe convective wind events and favourable environments is less correlated in the mid-century scenario compared with the historical simulation.

These findings are intended to provide insight on future regional projections in the frequency and intensity of severe wind gusts associated with convection. Future work should continue to evaluate convection-permitting models and their representa-

tion of convective hazards, with an emphasis on physical processes. In addition, experiments using forcings from a range of future climate scenarios should provide important information for projections of severe convective hazards, including an estimation of scenario uncertainty that has not been investigated here. Results show that BARPAC-M appears somewhat suitable
475 in the application of future wind projections, based on the ability to reliably represent the extreme tail of the distribution of daily maximum wind gusts and SCW characteristics aggregated over a large number of events. However, the overestimation of simulated events in steep lapse rate environments strongly influences the future changes indicated by the model. This change disagrees with estimates based on environmental changes, and represents a significant uncertainty. This suggests that future projections for convective hazards such as SCWs should be done using a multi-model approach, to account for uncertainties in
480 the representation of convective processes.

*Code and data availability.* The data used in this manuscript is available at https://doi.org/10.5281/zenodo.10521068 (Brown et al., 2024), except for WWLLN data that is commercial (see https://wwlln.net/). Analysis scripts and notebooks are available at https://github.com/andrewbrown31/BARPA/tree/main.

*Author contributions.* AB, AD and TL were responsible for conceptualizing the research and developing the methods. AD and TL provided
485 data, computational resources and supervision. AB carried out the formal analysis and wrote the original draft, while AD and TL assisted with reviewing and editing.

*Competing interests.* The authors declare that they have no conflict of interest.

*Acknowledgements.* This research was supported by the Australian Research Council's (ARC) Centre of Excellence for Climate Extremes (CE170100023). Computational support and resources were provided by the Australian National Computational Infrastructure. BARPA
490 data were produced and provided by the Australian Bureau of Meteorology as part of the Electricity Sector Climate Information project (https://www.climatechangeinaustralia.gov.au/en/projects/esci/), and the authors would like to personally thank Christian Stassen for assisting with accessing that data. We would also like to thank Andreas F. Prein and an anomymous reviewer for their comments and suggestions as part of the peer review process. The analysis here utilised several python software packages, including wrf-python (Ladwig, 2017), xesmf (Zhuang et al., 2023), sharppy (Blumberg et al., 2017) and seaborn (Waskom, 2021).

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
