# Peer review of "Convection-permitting climate model representation of severe convective wind gusts and future changes in southeastern Australia"

_EGUsphere, 2024_

## Author Comment (AC1)

**Response to Reviewer Comment 1 for Convection-permitting climate model representation of severe convective wind gusts and future changes in southeastern Australia**

**Reviewer comment 1**

The manuscript "Convection-permitting climate model representation of severe convective wind gusts and future changes in southeastern Australia" by Brown et al. investigates how future convective wind gusts might change over Southeastern Australia. Previous research suggests that surface hazards from thunderstorms, like severe convective winds (SCWs), may alter with climate change, yet global climate models struggle to resolve these due to their small scale, leading to uncertain projections. The authors find that SCW events using a convection-permitting climate model (BARPAC-M) over southeastern Australia for December–February, and comparing with a regional parent model (BARPAR) improved representation of extreme wind gusts in BARPAC-M but overestimation of SCW frequency, particularly in certain environments. Projected changes in SCW frequency for 2050 show uncertainties, with potential decreases under certain conditions and increases under others, highlighting the complexity of future SCW trends in the region. The study is very well structured, and written, and the images and text are of high quality. The differentiation of wind gusts into different categories adds a lot of novel insights about model biases and future climate change impacts on these extremes. This is one of the best-written and interesting papers that I have read in a while. I have only a couple of minor suggestions for changes and recommend publishing this manuscript after those are addressed.

Thank you for this review and providing comments and suggestions on the manuscript, including the positive feedback on our approach. We will respond to your comments below.

**General comment**

Adding more discussion on the importance of climate internal variability on your results would be beneficial. You mention the high degree of spatial and temporal variability of convective gusts already but making the role of internal climate variability more explicit would be important (see e.g., Deser et al. 2012). Internal climate variability could easily be the dominant source of uncertainty in your future climate projections.

Deser, C., Phillips, A., Bourdette, V. and Teng, H., 2012. Uncertainty in climate change projections: the role of internal variability. Climate dynamics, 38, pp.527-546.

We agree with the reviewer that the role of internal climate variability should be

mentioned, especially in the context of high spatial and temporal variability of severe convective wind events that is discussed in the submitted manuscript. We have included a paragraph on this in the Discussion of the revised manuscript, as shown below, that is also aimed to address similar comments from another reviewer:

*"The uncertainties related to extreme wind events mentioned above could also be associated with internal climate variability, including natural modes of variability such as the El Nino-Southern Oscillation (ENSO) and others, that could potentially affect the future projections presented here more generally. Internal variability can be a source of uncertainties in future climate projections (Deser et al., 2012) as well as in historical trends of severe thunderstorm environments for this region (Allen and Karoly 2014). Uncertainties from internal variability could be exacerbated by the relatively short 20-year period used for the analysis here, noting that this was the maximum period of data available for use and that these data were used in previous research such as described in Dowdy et al. (2019). Simulating long periods with convection-permitting models is very computationally demanding, as noted by previous studies that have used temporal windows of about 10-15 years length for future projections of severe convection (Gensini and Mote, 2014; Ashley et al., 2023). Future work to provide convection-permitting climate model simulations over longer periods will be beneficial, including with a larger sample helping to reduce the influence of internal climate variability (e.g., associated with ENSO) on estimates of longer-term climate changes. In addition, the relationships between SCW events and individual modes of climate variability in this region are relatively uncertain, with conflicting results for severe convection based on lightning and hail observations, and severe thunderstorm environments (Allen and Karoly, 2014; Dowdy, 2016; Soderholm et al., 2017; Dowdy 2020). Future work towards revealing these relationships could likely provide additional insights on the potential impact of internal climate variability on historical and future trends in convective hazards, including severe wind gusts"*.

Allen, J. T. and Karoly, D. J.: A climatology of Australian severe thunderstorm environments 1979 – 2011 : Inter-annual variability and ENSO influence, International Journal of Climatology, 34, 81–97, https://doi.org/10.1002/joc.3667, 2014

Ashley, W. S., Haberlie, A. M., and Gensini, V. A.: The Future of Supercells in the United States, Bulletin of the American Meteorological Society, 104, E1–E21, https://doi.org/10.1175/BAMS-D-22-0027.1, 2023.

Dowdy, A.: Seasonal forecasting of lightning and thunderstorm activity in tropical and temperate regions of the world, Scientific Reports, 6, 1–10, https://doi.org/10.1038/srep20874, 2016

Dowdy, A., Brown, A., Pepler, A., Thatcher, M., Rafter, T., Evans, J., Ye, H., Su, C.-H., Bell, S., Stassen, C. (2021). Extreme temperature, wind and bushfire weather projections using a standardised method. In Bureau Research Report – BRR055. http://www.bom.gov.au/research/publications/researchreports/BRR-055.pdf

Dowdy, A.J., 2020. Climatology of thunderstorms, convective rainfall and dry lightning environments in Australia. *Climate Dynamics*, *54*(5), 3041-3052, https://doi.org/10.1007/s00382-020-05167-9 .

Gensini, V. A. and Mote, T. L.: Estimations of hazardous convective weather in the United States using dynamical downscaling, Journal of Climate, 27, 6581–6589, https://doi.org/10.1175/JCLI-D-13-00777.1, 2014.

Soderholm, J., McGowan, H., Richter, H., Walsh, K., Weckwerth, T. M., and Coleman, M.: An 18-year climatology of hailstorm trends and related drivers across southeast Queensland, Australia, Quarterly Journal of the Royal Meteorological Society, 143, 1123–1135, https://doi.org/10.1002/qj.2995, 2017.

**Specific Comments**

1. Is BARPAC-M online or offline nested into BARPAR? The online nesting would have the benefit of providing higher-temporal resolution at the lateral boundaries which generally reduces that spatial spinup in the high-resolution domain.
   Thank you for providing this interesting point. BARPAC-M was run offline, with the revised manuscript text now noting the following "The convection-permitting model used here is a part of the broader BARPA modelling framework from the Australian Bureau of Meteorology (Su et al., 2021) ... The BARPAC-M simulations were run after the full time period of BARPA-R simulations were completed. BARPAC-M can therefore be considered an offline simulation with no feedback into the BARPA-R simulation". We will also pass this comment on to the Bureau of Meteorology modelling group that are running the BARPA simulations, in case they might consider online nesting for their future planning of convection-permitting simulations, noting the benefits mentioned in this review comment.

2. How did you account for boundary effects in BARPAC-M? Do you use a sponge zone and did you exclude boundary grid cells from the analysis?
   Yes it used a sponge zone, with that zone excluded from the analysis. The revised manuscript text now notes the following "The BARPAC-M simulations use a sponge zone for the lateral boundary nesting in the BARPA-R simulations, with analysis excluding data from that sponge zone, as was also the case for the BARPA-R simulations nested in ERA-Interim (Su et al 2021)."

3. L100: It is optimistic to assume that the resolved gust in BARPAR is 10-min if you have a 5-min time step. I would assume that your resolved temporal scales are at least 4Dt based on numerical considerations and model diffusivity.
   In the text referenced by the reviewer, it was not our intention to infer that BARPA-R can resolve gust processes on 10-minute scales. Instead, we meant to explain that the difference in output between BARPAC-M (maximum 3-second wind gust over all time steps in a 10-minute period) and BARPA-R (instantaneous 3-second wind gust at 10-minute intervals) should not impact the comparison between the two model configurations, given that the BARPA-R time step is relatively coarse compared with the 10-minute output frequency. We have made this clearer by including the following in the revised manuscript:

   *"Parameterised 3-second wind gust output is saved at 10-minute intervals in both models. For BARPAC-M, the 10-minute output represents the maximum 3-second gust over all model time steps. In contrast, for BARPA-R, the 10-minute output represents the wind gust from a single model time step, based on a model time step of 5 minutes. For the purposes of comparing the two model configurations, the BARPA-R 3-second gust distribution based on instantaneous*

*10-minute output is expected to be similar to a 10-minute maximum, given the relatively coarse model time step of 5 minutes, but with slightly lower mean and extreme values. This is demonstrated in Supplementary Material (Section S1) by resampling observational wind gust data to 5-minute intervals, where a bias of around -1.0 to -0.5 m/s is introduced using 10-minute instantaneous observed gusts relative to 10-minute maximum observed gusts. However, this bias is not expected to significantly impact the analysis of extreme wind gusts associated with convection, where spatial resolution and physical process representation are most relevant when comparing between models."*

4. In the analysis of Fig. 2 you directly compare the grid cell wind gust from the models with observed gusts at point locations. Should the model be able to capture point-scale wind gusts? I would assume that the model wind speed should be lower than that observed at point locations since the model is representing a spatial (e.g., grid cell) average wind gust, which in case of ERA5 and BARPAR is a quite large area.

This is a good point raised by the reviewer, and is not one that we address directly in the manuscript (although we do re-size the BARPAC-M grid to investigate the impact of grid cell size). We have now mentioned this as a disclaimer in the manuscript when introducing the analysis of Figure 2:

*"Here we compare the wind gust intensity distribution between each of the BARPA hindcast model configurations, the forcing model (ERA-Interim), and observations measured from AWS. ... It should be noted that the observations used here are representative of a single point location, compared with each of the model datasets that are intended to represent a grid cell average. Therefore, some differences between the observed wind gust distribution and model distributions should be expected, including lower model wind speeds for local wind gust events at station locations in general. The effect of model grid spacing will be investigated later in this section for BARPAC-M".*

And when discussing the results for ERA-Interim:

*"ERA-Interim is shown to realistically represent wind gust percentiles up to around 15 m/s, while significantly underestimating percentiles above this. This is likely due to the large grid cell area of this model dataset, as mentioned above in relation to comparing with point observations".*

5. 3: Maybe using a log y-axis would make this figure easier to read.
We agree, and have changed the y-axis to a log scale. Thank you for the suggestion.

6. L223: Why are you using the 6 km speed here? Are you assuming that this is the source height of downdrafts?
We use the 0-6 km mean wind speed as an estimate of the background wind that is likely relevant for storm motion and horizontal momentum available for downdrafts, although downdrafts might initiate much lower than this height in

some cases. We also already use this data for the event clustering, so it is readily available for the analysis of the ratio of the peak surface wind gust to the background wind speed. We have inserted the following text into the revised manuscript to communicate this:

*"A 0-6 km layer is chosen as this is likely representative of the background flow relevant for vertical mixing by downdrafts and storm motion, with wind speed data over this layer already available based on its application for event clustering (see Supplementary Material Section S2)".*

7. 6: Please add a legend that describes the circle sizes.
   We have added a legend. Thank you for the suggestion.

---

## Author Comment (AC2)

**Response to Reviewer Comment 3 for Convection-permitting climate model representation of severe convective wind gusts and future changes in southeastern Australia**

**Reviewer comment 3**

The authors investigated the severe convective wind (SCW) events based on the climate simulations over eastern Australia using the regional climate model at convective permitting resolution (BARPAC-M) to resolve the deep convections and coarse resolution (BARPA-R) to represent the environmental conditions. The work demonstrated the improved simulation of SCW events with convection-permitting resolution and investigated the SCW in different categories and their possible changes by the middle of the 21st century under the SSP585 scenario. The manuscript is very well written. The design of the experiments and methodology is sound at the bulk part, and the clustering-based analysis is very interesting! I would recommend minor changes before it is accepted for publication.

Thank you very much for your review and comments, we have responded to each comment below on a point-by-point basis.

**General comments**

First, there is a concern about the direct comparison of a given cluster between observation and simulation. The clusters from the k-mean method represent the relative differences in the same dataset. Although the partitioning of the events (e.g. percentage of each category) is comparable between the observation and simulation, the direct comparison of each cluster between observation and simulation may not be reasonable since a given cluster may not be physically similar enough between observation and simulation.

We thank the reviewer for this comment and acknowledge a range of uncertainties relating to the application of methods trained on observations then applied to model data. Although many studies use similar approaches of training methods to observations before applying them to model data, we believe it is important to understand details as much as possible for a given study approach. As such, the following analysis is provided to help demonstrate the suitability of applying these methods to model data.

Firstly, we have investigated this point further by visualising the k-means clustering following Brown et al. (2023), for both the observed and modelled (BARPAC-M) severe convective wind datasets, using the four clustering variables (see Supplementary Material Section S2 for further details):

- Mean water vapour mixing ratio in the lowest 1 km (Qmean01)

- Vertical wind shear between the surface and 6 km (S06)
- The temperature lapse rate between 1 and 3 km (LR13)
- The mass-weighted mean wind speed from the surface to 6 km (Umean06)

This visualisation is shown below, for both the observed SCW event dataset (using clustering variables from ERA5), and the simulated BARPAC-M event dataset (using clustering variables from BARPA-R). The below figure demonstrates that the distribution of those variables in each SCW cluster is similar between ERA5 and BARPA-R, that we believe gives some confidence in the application of the clustering method to the BARPA dataset (although there are some differences such as less spread in the BARPA-R distribution across SCW events for Umean06 and LR13, and many more steep lapse rate SCW events simulated by BARPAC-M as noted in the main manuscript text). This comparison will be discussed in the revised version of the manuscript (Section 3.3), while the figure and additional discussion is shown in the Supplementary Material (along with more general evaluation of BARPA-R diagnostics compared with ERA5, in Section S2).

[Figure]

*Joint distributions of (left column) the mean water vapour mixing ratio in the lowest 1 km (Qmean01) and vertical wind shear between the surface and 6 km (S06), as well as (right column) the*

*temperature lapse rate between 1 and 3 km (LR13) and the mass-weighted mean wind speed from the surface to 6 km (Umean06), for severe convective wind (SCW) events from (top row) observations and (bottom row) BARPAC-M. For observed SCW events, the environmental diagnostics shown here are calculated from ERA5, while for BARPAC-M events, the diagnostics are calculated from BARPA-R. The joint distributions are separated based on the clustering method of Brown et al. (2023), including (red) strong background wind, (yellow) steep lapse rate, and (blue) high moisture SCW events.*

Secondly, we believe that the separation of physical SCW characteristics between event types is replicated reasonably well by BARPAC-M when compared with observations (see Figure 5 of the submitted manuscript), thereby helping to provide confidence in the applicability of the observations-based clustering method to the BARPA model data. We have provided some additional discussion on this in the revised manuscript (Section 3.3), as follows, with relevant points for this reviewer comment shown below in bold text:

*"Consistent with Brown et al. (2023), Figures 5c, e, and g demonstrate that the observed wind gust ratio, daily lightning flashes, and deep-layer wind ratio all increase with clusters that are supportive of relatively deep convection. That is, these quantities are highest for SCW events in the high moisture cluster, followed by the steep lapse rate and strong background wind clusters.* **BARPAC-M is able to replicate this behaviour for the distribution of daily lightning flashes (Figures 5f) and the deep-layer wind ratio (Figures 5h). This broad separation of SCW characteristics between different types of events gives some confidence in the suitability of applying the clustering method to BARPAC-M. In addition, the distribution of large-scale environmental diagnostics is broadly consistent between observations and BARPAC-M across event types, as shown in the Supplementary Material (Section S2).** *However, there are also some key differences in SCW characteristics between BARPAC-M and the observed dataset, for different event types. For example, while, BARPAC-M produces higher wind gust ratios for SCWs associated with the steep lapse rate cluster compared with the strong background wind cluster, as observed (Figure 5d), the model produces significantly lower values for the high moisture cluster compared with observations. This suggests that key processes related to SCWs within the high moisture cluster are not represented in BARPAC-M. This could relate to, for example, supercell downdrafts that occur on small spatial scales, and mostly occur within this cluster (Brown et al., 2023). There appears to be little difference in the severe wind gust intensity distribution between clusters, based on the results here (Figures 5a–b)".*

Brown, A., Dowdy, A., Lane, T. P., and Hitchcock, S.: Types of Severe Convective Wind Events in Eastern Australia, Monthly Weather Review, 151, 419–448, https://doi.org/10.1175/MWR-D-22-0096.1, 2023

Another concern is that the 20-year simulation may not cover long enough climate variability, thus the future changes are sensitive to the resampling. It might be beneficial to show the phases of the major climate variability are similar between the historical and future runs.

We agree that this is a source of uncertainty and have added a new paragraph to the revised manuscript in the Discussion section. The new paragraph provides details in response to this review comment as well as to a similar comment from Reviewer 1 (as detailed in the responses to Reviewer 1).

This issue on the impact of natural climate variability for future projections is commonly discussed within climate model studies. This is especially the case for studies that use convection-permitting models, since the large amounts of computational resources needed for these types of simulations generally results in relatively short analysis periods. The 20-year window used here for these simulations provided by the Bureau of Meteorology is consistent with the current capability of large modelling centres, and is a similar length to previous studies for thunderstorm projections (Gensini and Mote 2014, Ashley et al. 2023).

As noted in the new paragraph added, we agree that the 20-year window does lead to the potential for natural climate variability to not be fully sampled in a balanced way, such as due to the potential for some influence from large-scale atmospheric and oceanic modes of variability such as ENSO and many others. We have chosen to not calculate the phase of individual climate modes in the historical and future runs because robust relationships between severe convection and individual climate modes have not been found to date for this region, noting conflicting results on this from previous studies. This is now discussed in the new paragraph to be inserted in the revised Discussion, as follows:

*"The uncertainties related to extreme wind events mentioned above could also be associated with internal climate variability, including natural modes of variability such as the El Nino-Southern Oscillation (ENSO) and others, that could potentially affect the future projections presented here more generally. Internal variability can be a source of uncertainties in future climate projections (Deser et al., 2012) as well as in historical trends of severe thunderstorm environments for this region (Allen and Karoly 2014). Uncertainties from internal variability could be exacerbated by the relatively short 20-year period used for the analysis here, noting that this was the maximum period of data available for use and that these data were used in previous research such as described in Dowdy et al. (2019). Simulating long periods with convection-permitting models is very computationally demanding, as noted by previous studies that have used temporal windows of about 10-15 years length for future projections of severe convection (Gensini and Mote, 2014; Ashley et al., 2023). Future work to provide convection-permitting climate model simulations over longer periods will be beneficial, including with a larger sample helping to reduce the influence of internal climate variability (e.g., associated with ENSO) on estimates of longer-term climate changes. In addition, the relationships between SCW events and individual modes of climate variability in this region are relatively uncertain, with conflicting results for severe convection based on lightning and hail observations, and severe thunderstorm environments (Allen and Karoly, 2014; Dowdy, 2016; Soderholm et al., 2017; Dowdy 2020). Future work towards revealing these relationships could likely provide additional insights on the potential impact of internal climate variability on historical and future trends in convective hazards, including severe wind gusts".*

Allen, J. T. and Karoly, D. J.: A climatology of Australian severe thunderstorm environments 1979 – 2011 : Inter-annual variability and ENSO influence, International Journal of Climatology, 34, 81–97, https://doi.org/10.1002/joc.3667, 2014

Ashley, W. S., Haberlie, A. M., and Gensini, V. A.: The Future of Supercells in the United States, Bulletin of the American Meteorological Society, 104, E1–E21, https://doi.org/10.1175/BAMS-D-22-0027.1, 2023.

Dowdy, A.: Seasonal forecasting of lightning and thunderstorm activity in tropical and temperate regions of the world, Scientific Reports, 6, 1–10, https://doi.org/10.1038/srep20874, 2016

Dowdy, A., Brown, A., Pepler, A., Thatcher, M., Rafter, T., Evans, J., Ye, H., Su, C.-H., Bell, S., Stassen, C. (2021). Extreme temperature, wind and bushfire weather projections using a standardised method. In Bureau Research Report – BRR055.
http://www.bom.gov.au/research/publications/researchreports/BRR-055.pdf

Dowdy, A.J., 2020. Climatology of thunderstorms, convective rainfall and dry lightning environments in Australia. *Climate Dynamics*, *54*(5), 3041-3052, https://doi.org/10.1007/s00382-020-05167-9 .

Gensini, V. A. and Mote, T. L.: Estimations of hazardous convective weather in the United States using dynamical downscaling, Journal of Climate, 27, 6581–6589, https://doi.org/10.1175/JCLI-D-13-00777.1, 2014.

Soderholm, J., McGowan, H., Richter, H., Walsh, K., Weckwerth, T. M., and Coleman, M.: An 18-year climatology of hailstorm trends and related drivers across southeast Queensland, Australia, Quarterly Journal of the Royal Meteorological Society, 143, 1123–1135, https://doi.org/10.1002/qj.2995, 2017.

**Specific comments**

1. Line 30-31: The changes in large-scale environment affecting deep convections may not be limited to temperature and moisture. Other factors may be worth noticing/mentioning, such as changes in tropospheric lapse rate, low-level wind shear, and relative humidity.
Thank you for pointing this out, we now mention these other factors in the revised manuscript that may be relevant for future changes in deep convection in the Introduction, as follows:

   "Future human-induced climate change could also potentially impact other factors relevant for deep moist convection, such as temperature lapse rates, vertical wind shear, and relative humidity (Seeley and Romps, 2015; Chen et al., 2020)".

   Chen, J., Dai, A., Zhang, Y., and Rasmussen, K. L.: Changes in Convective Available Potential Energy and Convective Inhibition under Global Warming, Journal of Climate, 33, 2025–2050, https://doi.org/10.1175/JCLI-D-19-0461.1, 2020

   Seeley, J. T. and Romps, D. M.: The effect of global warming on severe thunderstorms in the United States, Journal of Climate, 28, 2443–2458, https://doi.org/10.1175/JCLI-D-14-00382.1, 2015

2. Line 54-64: The discussion on model uncertainties touches only on the influence of model resolution on dynamics. It might be worth mentioning other sources of

uncertainty although resolution is the focus here. For example, the uncertainties in parameterizations, especially the microphysical (MP) parametrization may be worth mentioning since the deep convective systems and growth of hydrometers are strongly affected by MP processes, which further affect the evaporative cooling and downdraft.

We provide some additional discussion of uncertainties related to various unresolved dynamics in convection-permitting models, including MP processes. The following is included in the Introduction of the revised manuscript:

*"...These errors are likely related to several dynamical processes that are not sufficiently resolved by these model configurations, such as microphysical processes, convective cold pools, and entrainment of environmental air (Bryan and Morrison, 2012; Jucker et al., 2020; Bergemann et al., 2022). Errors in convective storm timing and location have also been shown to be caused by deficiencies in boundary conditions in some cases, relating to representations of the large-scale thermodynamic environment (Hanley and Lean, 2021). Also, the turbulent nature of (severe convective wind) events means that the relevant surface winds are often parameterised in these models (Hawbecker, 2020), which can lead to biases in surface wind speed"*

Bergemann, M., Lane, T. P., Wales, S., Narsey, S., and Louf, V.: High-resolution simulations of tropical island thunderstorms: Does an increase in resolution improve the representation of extreme rainfall?, Quarterly Journal of the Royal Meteorological Society, 148, 3303–3318, https://doi.org/10.1002/qj.4360, 2022

Bryan, G. H. and Morrison, H.: Sensitivity of a simulated squall line to horizontal resolution and parameterization of microphysics, Monthly Weather Review, 140, 202–225, https://doi.org/10.1175/MWR-D-11-00046.1, 2012

Hanley, K. E. and Lean, H. W.: Elucidating the causes of errors in 2.2 km Met Office Unified Model simulations of a convective case over the US Great Plains, Quarterly Journal of the Royal Meteorological Society, pp. 1–19, https://doi.org/10.1002/qj.4049, 2021

Jucker, M., Lane, T. P., Vincent, C. L., Webster, S., Wales, S. A., and Louf, V.: Locally forced convection in subkilometre-scale simulations with the Unified Model and WRF, Quarterly Journal of the Royal Meteorological Society, 146, 3450–3465, https://doi.org/10.1002/qj.3855, 2020

3. Line 90: Climatology based on the "20-year" simulation may be sensitive to climate variability depending on when the period starts. It may be beneficial to show the historical and future periods cover similar phases of the major climate modes that affect deep convection over eastern Australia.
   See response to general comment, where we also note that there are considerable uncertainties in the relationships between deep convection over eastern Australia and major climate modes.

4. Line 132-136: It might be easier for the readers to follow if the introduction of the terms is in the same sequence as their occurrence in Eq. 2.
   We have made this change in the revised manuscript, thank you.

5. Line 243-246: Since k-mean clustering captures the relative difference within the same data, is it possible that the "severe high moisture events" from the observation and simulation are not physically similar enough to make a direct comparison? It would be better to show some evidence of how similar the two clusters are in terms of moisture and related properties before we conclude here. Same suggestion for other clusters if compared directly between observation and simulation.
   See response to general comments about clustering.

6. Figure 7: Since both BARPAC-M and BARPA-R are presented in each panel, It is better to use a common y-axis title to describe both simulations.
   Thanks for this suggestion, we have replaced the y-axis title with a common "Model" label.

7. Line 258-259: I don't see a causality relation between the bias in intensity distribution and an overestimation of event numbers here
   Because of the high-bias in intensity, a greater proportion of the BARPAC-M wind gust distribution lies above 25 m/s, which is the threshold used for severe convective wind events in our study. Therefore, this result can be related to the bias in event frequency discussed earlier in the manuscript. We have clarified this in the revised manuscript with the following text:

   *"Results indicate that BARPAC-M tends to overestimate the intensity distribution of convective daily maximum wind gusts (Figure 7d–f), especially when considering severe gusts over 25 m/s in the steep lapse rate cluster (Figure 7e). This bias results in a greater portion of the BARPAC-M convective wind gust distribution exceeding the 25 m/s severe threshold in steep lapse rate environments, compared with observations, and therefore relates to the overestimation of simulated steep lapse rate SCW events presented earlier (Figure 6c and g) and in the overall number of SCW events (Figure 6a and e)".*

8. Line 261-262: Based on Figure S7, the overestimated amount of steep lapse rate SCW events does show consistent bias to the large-scale environment from BARPA-R that the model produces more frequent "steep lapse rate" conditions than observed.
   Thanks for pointing this out, we amended the text here based on the reviewer's comment. We conclude that the overestimated amount of steep lapse rate SCW events might somewhat be due to bias in the large-scale environment, but that this bias is relatively small compared with the bias in SCW frequency. We included the updated paragraph in the revised manuscript:

   *"This overestimation in the amount of steep lapse rate SCW events could be due to several factors. This includes biases in the large-scale environment inherited from BARPA-R, with a slightly higher occurrence frequency of favourable steep lapse rate environments compared with ERA5 (see Supplementary Material Figure S7b). However, the relative bias in favourable steep lapse rate environments is much smaller than the relative bias in simulated SCW event*

*frequency from BARPAC-M within steep lapse rate environments (compare Supplementary Material Figures S7b and c). This suggests that the bias in simulated SCW event frequency is not primarily driven by biases from BARPA-R, and is instead due to errors in dynamical processes related to SCWs in BARPAC-M, such as convective downdrafts that are too intense or numerous, or errors in gust parameterisations (explored further in Discussion section). This is further supported by a relatively consistent frequency in daily lightning occurrence between BARPAC-M and WWLLN for steep lapse rate environments, compared with the frequency of SCW occurrence (see Supplementary Material Figure S7d)".*

9. Line 271-272: This uncertainty may be due to uncertainty in climate variability. See response to general comment about internal climate variability.

10. Line 300: My guess is the BDSD and criteria used to identify the F_ENV were developed based on a different dataset other than the model simulations. It is worth considering 1) the uncertainty in this parameterized metric when applied to a different dataset, and 2) whether the model can well capture the physical relations between the parameters considered in BDSD and SCW. It would be beneficial if some physical explanations could be provided for the opposite changes in SCW and F-ENV.
This is a good point, and it is correct that the BDSD and F_ENV criteria were trained on a different dataset to the model simulations (it was trained on an observational SCW dataset and the ERA5 reanalysis, see Brown and Dowdy (2021)). We have already indirectly addressed the uncertainties related to the application of the BDSD to future climate projections, on line 372 of the submitted manuscript:

*"The lack of correlation between future simulated SCWs and favourable SCW environments found here highlights potential uncertainties in the application of methods that have been developed in the historical climate, as also mentioned by previous studies (Hoogewind et al., 2017; Raupach et al., 2021). … the future changes reported here therefore have low confidence…".*

In response to the reviewer comment, we expand on this existing point in the revised manuscript, with the following text included:

*"The lack of correlation between future simulated SCWs and favourable SCW environments found here highlights potential uncertainties in the application of methods that have been developed in the historical climate, as also mentioned by previous studies (Hoogewind et al., 2017; Raupach et al., 2021). **This includes the application of the Brown and Dowdy (2021b) Statistical Diagnostic (BDSD) that is used here to diagnose favourable SCW environments (see Section 2.3). Additional uncertainty is also potentially introduced here by the fact that the BDSD was initially trained on an observational dataset (Brown and Dowdy, 2021b), and is being applied here to a regional climate model with a potentially different relationship between***

*SCW events and the large-scale environment. Future work could be aimed at further investigating the physical mechanisms relating favourable SCW environments to simulated events in current and future climates. Although the future changes reported here have low confidence based on this discussion, our results suggest a potential increase in the frequency of SCWs in high moisture environments, that may be compensated by a reduction in SCWs in steep lapse rate environments".*

In terms of the reviewer's suggestion that "It would be beneficial if some physical explanations could be provided for the opposite changes in SCW and F-ENV", we believe that this has already been somewhat investigated by the clustering method used throughout, including the analysis of future changes in steep lapse rate and high moisture SCW events. For example, on line 355 of the submitted manuscript, it is stated that:

*"Because the majority of BARPAC-M events occur in steep lapse rate environments, the overall future change suggested by that model is a decrease in simulated SCW frequency, while the opposite is true for the BARPA-R environmental method where high moisture environments are relatively important".*

However, we agree that this is an important point suggested by the reviewer and should be further investigated by future work, as noted in the above revised text. This is noted in the revised text as follows:

*"Future work could be aimed at further investigating the physical mechanisms relating favourable SCW environments to simulated events in current and future climates"*

Brown, A. and Dowdy, A.: Severe Convective Wind Environments and Future Projected Changes in Australia, Journal of Geophysical Research: Atmospheres, 126, 1–17, https://doi.org/10.1029/2021JD034633, 2021a.

Hoogewind, K. A., Baldwin, M. E., and Trapp, R. J.: The impact of climate change on hazardous convective weather in the United States: Insight from high-resolution dynamical downscaling, Journal of Climate, 30, 10 081–10 100, https://doi.org/10.1175/JCLI-D-16-0885.1, 2017

Raupach, T. H., Martius, O., Allen, J. T., Kunz, M., Lasher-Trapp, S., Mohr, S., Rasmussen, K. L., Trapp, R. J., and Zhang, Q.: The effects of climate change on hailstorms, Nature Reviews Earth & Environment, 2, 213–226, https://doi.org/10.1038/s43017-020-00133-9, 2021